# Live-cell single-molecule dynamics of PcG proteins imposed by the DIPG H3.3K27M mutation

Roubina Tatavosian[1], Huy Nguyen Duc[1], Thao Ngoc Huynh[1], Dong Fang[2], Benjamin Schmitt[3], Xiaodong Shi[4], Yiming Deng[4], Christopher Phiel[5], Tingting Yao[3], Zhiguo Zhang[2], Haobin Wang[1] & Xiaojun Ren[1]

Over 80% of diffuse intrinsic pontine gliomas (DIPGs) harbor a point mutation in histone H3.3 where lysine 27 is substituted with methionine (H3.3K27M); however, how the mutation affects kinetics and function of PcG proteins remains elusive. We demonstrate that H3.3K27M prolongs the residence time and search time of Ezh2, but has no effect on its fraction bound to chromatin. In contrast, H3.3K27M has no effect on the residence time of Cbx7, but prolongs its search time and decreases its fraction bound to chromatin. We show that increasing expression of *Cbx7* inhibits the proliferation of DIPG cells and prolongs its residence time. Our results highlight that the residence time of PcG proteins directly correlates with their functions and the search time of PcG proteins is critical for regulating their genomic occupancy. Together, our data provide mechanisms in which the cancer-causing histone mutation alters the binding and search dynamics of epigenetic complexes.

[1] Department of Chemistry, University of Colorado Denver, Denver, CO 80217-3364, USA. [2] Institute for Cancer Genetics, Columbia University, New York, NY 10032, USA. [3] Department of Biochemistry and Molecular Biology, Colorado State University, Fort Collins, CO 80523, USA. [4] Department of Electrical and Computer Engineering, Michigan State University, 428 S. Shaw Lane, East Lansing, MI 48824, USA. [5] Department of Integrative Biology, University of Colorado Denver, Denver, CO 80217-3364, USA. These authors contributed equally: Roubina Tatavosian, Huy Nguyen Duc, Thao Ngoc Huynh. Correspondence and requests for materials should be addressed to X.R. (email: xiaojun.ren@ucdenver.edu)

Epigenetic regulatory complexes play an essential role in the organization of chromatin structure, thereby modulating gene expression[1]. Polycomb group (PcG) proteins are well-characterized epigenetic regulators that are assembled into two distinct complexes, Polycomb repressive complex (PRC) 1 and PRC2[2]. PRC2 catalyzes trimethylation of histone H3 on lysine 27 (H3K27me3) via the catalytic subunit Ezh2 or Ezh1[3–7]. PRC1 complexes can ubiquitinate histone H2A at lysine 119 (H2AK119Ub) through their catalytic subunit Ring1a or Ring1b. Based on the protein subunit composition of these individual PRC1 complexes, they are divided into "canonical" or "variant" complexes[8,9]. Canonical PRC1 complexes (Cbx-PRC1; the functional homolog to *Drosophila* PRC1) assemble with either Pcgf2 (Mel18) or Pcgf4 (Bmi1), and incorporate a chromobox (Cbx) protein. PcG proteins play crucial roles during disease pathogenesis. Cbx7, one of the core components of Cbx-PRC1, and Ezh2 can be a proto-oncogene or a tumor suppressor in a context-dependent manner[10–15].

Diffuse intrinsic pontine gliomas (DIPGs) are aggressive primary brainstem tumors with a median age at diagnosis of 6–7 years and the leading cause of brain tumor-related death in children[16]. Recent genomic studies revealed that up to 80% of DIPG tumors exhibit a characteristic mutation of lysine 27 to methionine (K27M) in genes encoding histone H3.3 (*H3F3A*), and, to a lesser extent, H3.1 (*HIST3H1B*)[17–19]. H3K27M mutation occurs in a single allele of a total of 32 copies of histone H3 genes, and therefore only a minor fraction (3.6–17.6%) of the total H3 protein in DIPGs is the mutant[20]. Subsequent studies showed that the H3K27M mutation results in a global reduction of H3K27me3 levels[20–23]. However, the genome-wide sequencing revealed a striking retainment of H3K27me3 and PRC2 at selected genomic loci in H3K27M-mutant DIPGs[22,23]. How H3K27M reduces global H3K27me3 levels, but selectively retains H3K27me3 at a subset of genes in vivo is not yet fully understood.

Biochemical investigations identified that H3K27M peptides bind to the PRC2 active site with above 20-fold higher affinity than the equivalent peptides, H3K27M-mutant peptides or nucleosomes inhibit the PRC2 activity[20,24,25], and Ezh2 is enriched on H3K27M-containing mononucleosomes isolated from cells[20,22]. Based upon these classical biochemical studies[20,22,24,25] and the genome-wide sequencing[22,23], the sequestration model[24] and the inhibition model[23] have been proposed to explain the in vivo reprogramming of H3K27me3 by H3K27M. However, recent genome-wide studies showed that H3K27M localizes to transcriptionally active chromatin regions and at gene regulatory elements and PRC2 is largely excluded from chromatin on sites containing H3K27M[26], and both PRC2 and H3K27me3 are lost from weak Polycomb targets[23]. Another recent biochemical study demonstrated that PRC2 exhibits similar binding affinity for H3K27M-nucleosomes and unmodified nucleosomes[27]. These results are not fully compatible with the sequestration or inhibition model. Therefore, critical knowledge gaps remain in our understanding of the mechanistic links between the H3K27M mutant and PRC2 functionalities. Additionally, H3K27me3 is the mark for the targeting of Cbx7-PRC1 to chromatin[28]. Whether and how H3K27M affects the Cbx7-PRC1 targeting remains unexplored.

Direct measurement of the binding and search kinetics of PcG proteins in vivo is necessary for our understanding of the aforementioned mechanisms. Here, we utilize live-cell single-molecule tracking (SMT) and genetic engineering to determine the binding and search dynamics of PRC2 and Cbx7 and elucidate the kinetic mechanisms that underpin how H3.3K27M inhibits the PRC2 activity in vitro and reprograms the genomic occupancy of H3K27me3 and PRC2 in vivo. We further show that elevating expression of *Cbx7* inhibits the proliferation of DIPG cells and stabilizes Cbx7 on chromatin.

## Results

**PRC2 and Cbx7 have different chromatin-bound fractions**. To investigate the PRC2 binding dynamics at endogenous genomic loci within living cells, we generated mouse embryonic stem (mES) cells stably expressing HaloTag-PRC2 subunit fusions under the control of an inducible tetracycline response element-tight promoter. Unless otherwise indicated, we performed live-cell SMT experiments at the basal level of HaloTag-PRC2 subunit fusion expression without doxycycline induction. A small sub-population of HaloTag-PRC2 subunit fusion was labeled by bright and photostable Janelia Fluor 549 (JF$_{549}$)[29] and was illuminated using highly inclined thin illumination (HILO) mode (Fig. 1a)[30]. The number of fluorescently labeled HaloTag fusions within cells was at a range of 5–20 particles per frame (Fig. 1b).

We used H2A-HaloTag and HaloTag-NLS (NLS, nuclear localization sequence) to validate our live-cell SMT system[28]. Please note that in some cases HaloTag is abbreviated to HT. A large population of H2A-HaloTag was stationary (Supplementary Movie 1) while nearly all of HaloTag-NLS were highly mobile (Supplementary Movie 2). We tracked individual molecules and constructed the displacement histogram and the cumulative distribution of displacements (Fig. 1c, d). To calculate kinetic fractions and diffusion constants, we carried out kinetic modeling of the measured displacements using Spot-On[31]. The displacement distribution for H2A-HaloTag was fitted to generate chromatin-bound ($F_1$ and $D_1$) and free diffusion ($F_3$ and $D_3$) (Fig. 1d). The displacement distribution for HaloTag-NLS was decomposed into chromatin-bound ($F_1$ and $D_1$), confined diffusion ($F_2$ and $D_2$), and fast diffusion ($F_3$ and $D_3$) (Fig. 1d). Approximately 82% of H2A-HaloTag molecules bound to chromatin (Fig. 1e and Supplementary Table 1), which is consistent with a previous report using fluorescence recovery after photobleaching (FRAP)[32]. In contrast, only ~8% of HaloTag-NLS molecules bound to chromatin (Fig. 1e and Supplementary Table 1). Among the Cbx family proteins, Cbx7 is the well-characterized effector of H3K27me3 generated by PRC2 in mES cells[28]; therefore, we quantified the kinetic fractions and diffusion constants of Cbx7 (Fig. 1c, d and Supplementary Movie 3). We found that ~40% of HaloTag-Cbx7 binds to chromatin, ~40% is in confined diffusion, and ~20% is in free diffusion (Fig. 1e and Supplementary Table 1).

Given that Ezh2 is the catalytic subunit of PRC2 and Eed is the core component of PRC2, we quantitatively measured the kinetic fractions and diffusion constants of Ezh2 and Eed in living mES cells (Fig. 1c–e and Supplementary Movies 4-5). We found that ~23% of HaloTag-Ezh2 binds to chromatin, ~23% is in confined diffusion, and ~54% is in free diffusion (Fig. 1e and Supplementary Table 1). Likewise, ~20% of HaloTag-Eed associates with chromatin, ~29% is confined diffusion, and ~51% is in free diffusion (Fig. 1e and Supplementary Table 1). Since the HaloTag-PRC2 subunit fusions and their endogenous counterparts co-exist within cells, the fusion proteins may be at disadvantage to compete with their endogenous counterparts for binding sites. To this end, we measured the kinetic fractions of HaloTag-Ezh2 and HaloTag-Eed in *Ezh2*$^{-/-}$ and *Eed*$^{-/-}$ mES cells, respectively. The kinetic fractions of HaloTag-Ezh2 and HaloTag-Eed in their corresponding knockout mES cells were comparable to those obtained from wild-type mES cells (Fig. 1c–e and Supplementary Table 1). These data suggest that HaloTag is unlikely to interfere with the competition between the fusion proteins and their corresponding endogenous counterparts. To investigate whether the HaloTag fusion influences the enzymatic

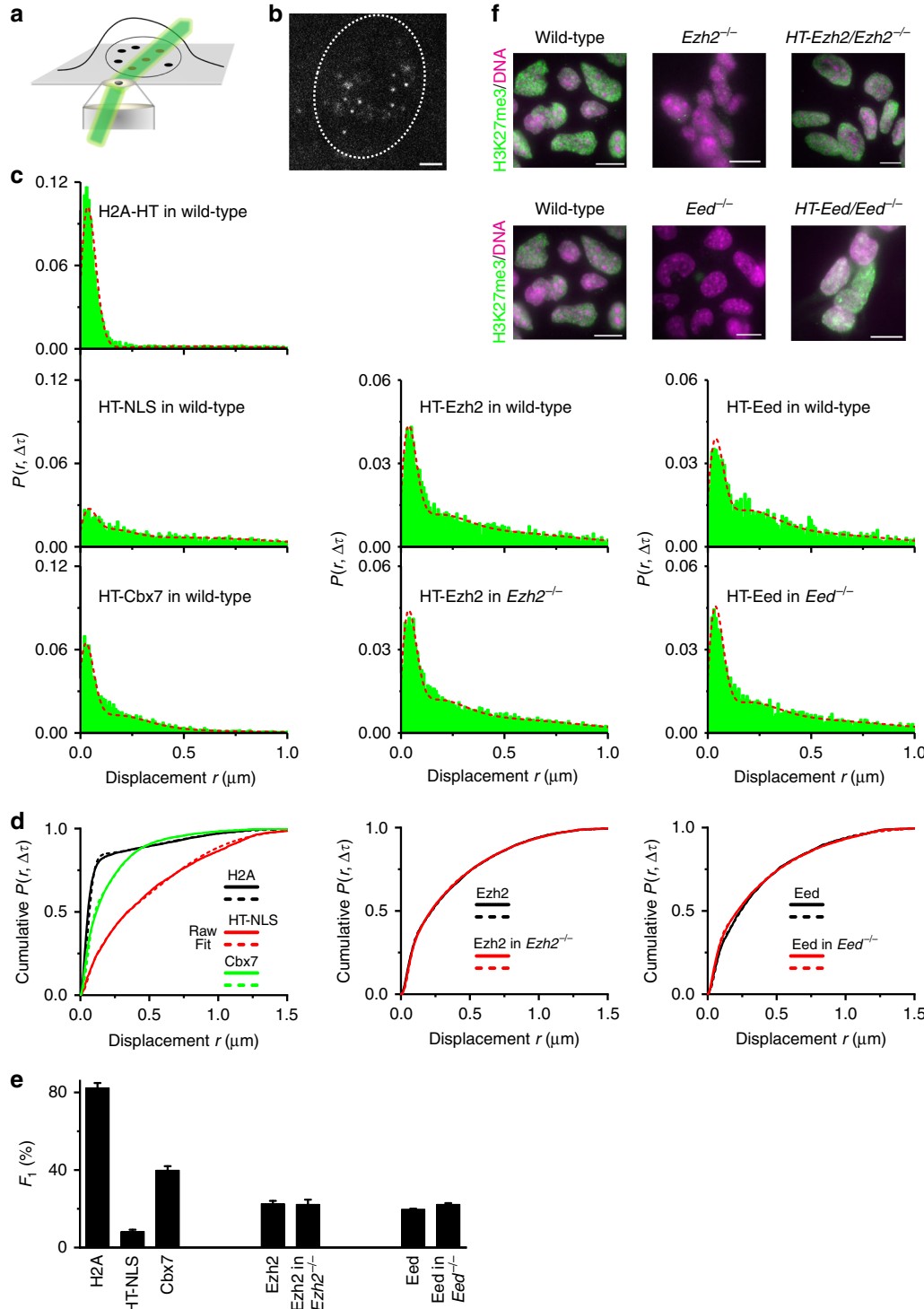

activity of PRC2, we quantified the H3K27me3 level in *HaloTag-Ezh2/Ezh2⁻/⁻*, *HaloTag-Eed/Eed⁻/⁻*, and wild-type mES cells by immunofluorescence. Our data indicated that the H3K27me3 level in *HaloTag-Ezh2/Ezh2⁻/⁻* and *HaloTag-Eed/Eed⁻/⁻* mES cells is similar to that in wild-type mES cells (Fig. 1f), suggesting that the fusion proteins biochemically recapitulate the function of their endogenous counterparts.

**PRC2 and Cbx7 have similar stability on chromatin.** The fraction of PcG bound to specific sites is determined by the ratio of off-rate (the inverse of residence time) and target search time

(Methods). The stability of PcG proteins on chromatin is determined by their residence time. To measure the residence time of Ezh2 and Eed on chromatin, we performed live-cell SMT at an integration time, $\tau_{int}$, of 30 ms interspersed with a dark time, $\tau_d$, of 170 ms as described previously (Fig. 2a)[28]. We considered molecules whose diffusion constant is less than 0.032 ($\mu m^2$ per s) as the chromatin-bound ones. This is a conservative consideration since we intend to avoid counting molecules of confined motion whose diffusion constant is at a range of 0.32–0.77 ($\mu m^2$ per s) (Supplementary Table 1). We counted the lifetime of the fluorescence spots as the dwell time of individual chromatin-

**Fig. 1** PRC2 and Cbx7 exhibit distinct capacities for binding to chromatin. **a** Schematic illustrating HILO (highly inclined and laminated optical sheet). **b** Example image showing single HaloTag-Ezh2 molecules labeled with JF$_{549}$ dye during a 30 ms exposure time. The nucleus was marked by oval white dash circle. The individual white points represent single HaloTag-Ezh2 molecules. Scale bar, 2.0 μm. **c** Displacement histograms for H2A-HaloTag ($N = 52$ cells, $n = 6172$ displacements), HaloTag-NLS ($N = 66$ cells, $n = 6587$ displacements), and HaloTag-Cbx7 ($N = 40$ cells, $n = 7725$ displacements) in wild-type mES cells, for HaloTag-Ezh2 in wild-type ($N = 30$ cells, $n = 5532$ displacements) and $Ezh2^{-/-}$ ($N = 50$ cells, $n = 14,427$ displacements) mES cells, and for HaloTag-Eed in wild-type ($N = 36$ cells, $n = 5444$ displacements) and $Eed^{-/-}$ ($N = 77$ cells, $n = 8845$ displacements) mES cells. The short dash red curve indicates the overall fit. NLS, nuclear localization sequence. HT, HaloTag. **d** Cumulative distribution of displacements for H2A-HaloTag, HaloTag-NLS, and HaloTag-Cbx7 in wild-type mES cells, for HaloTag-Ezh2 in wild-type and $Ezh2^{-/-}$ mES cells, and for HaloTag-Eed in wild-type and $Eed^{-/-}$ mES cells. The cumulative distributions were fitted with two or three components. Fitted parameters are shown in Supplementary Table 1. Unless otherwise indicated, the reported kinetic fractions and diffusion constants were obtained from the cumulative distributions. Solid curve represents raw data. Short dash curve is fitted data. **e** Fraction of the chromatin-bound population ($F_1$) obtained from **d**. Results are means ± SD from three biological replicates. **f** Immunostaining of H3K27me3 in wild-type, $Ezh2^{-/-}$, $Eed^{-/-}$, $HaloTag-Ezh2/Ezh2^{-/-}$, and $HaloTag-Eed/Eed^{-/-}$ mES cells by using antibody directed against H3K27me3 (green). DNA was stained with hoechst (red). Overlay images are shown. The residual H3K27me3 level was detectable in $Ezh2^{-/-}$ mES cells because of the presence of Ezh1. Scale bar, 5.0 μm

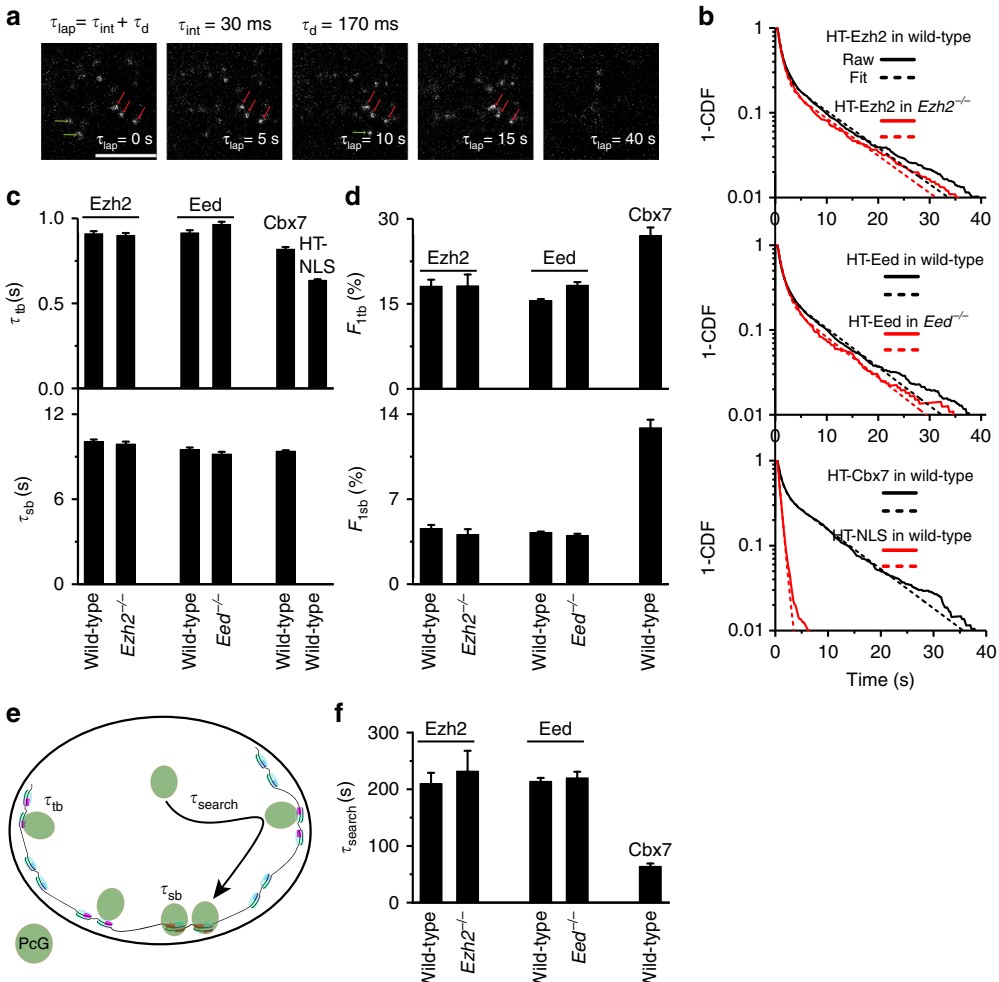

**Fig. 2** PRC2 and Cbx7 possess a similar residence time, but exhibit distinct search dynamics. **a** Example images showing single HaloTag-Ezh2 molecules binding to and unbinding from chromatin. The red arrowhead indicates the HaloTag-Ezh2 molecules bound to chromatin and the green arrowhead represents the HaloTag-Ezh2 molecules unbound from chromatin. Scale bar is 5.0 μm. **b** Survival probability distribution of the dwell times for HaloTag-Ezh2 in wild-type ($N = 112$ cells, $n = 4854$ trajectories) and $Ezh2^{-/-}$ ($N = 77$ cells, $n = 2365$ trajectories) mES cells, for HaloTag-Eed in wild-type ($N = 47$ cells, $n = 3533$ trajectories) and $Eed^{-/-}$ ($N = 56$ cells, $n = 1222$ trajectories) mES cells, and for HaloTag-Cbx7 ($N = 27$ cells, $n = 1924$ trajectories) and HaloTag-NLS ($N = 61$ cells, $n = 2973$ trajectories) in wild-type mES cells. The distributions were fitted with a two-component exponential decay model. Fitted parameters are shown in Supplementary Table 2. **c** Residence time of the short-lived ($\tau_{tb}$) and long-lived ($\tau_{sb}$) population for HaloTag-Ezh2 in wild-type and $Ezh2^{-/-}$ mES cells, for HaloTag-Eed in wild-type and $Eed^{-/-}$ mES cells, and for HaloTag-Cbx7 and HaloTag-NLS in wild-type mES cells. Results are means ± SD. **d** Fraction of the short-lived ($F_{1tb}$) and long-lived ($F_{1sb}$) population for HaloTag-Ezh2 in wild-type and $Ezh2^{-/-}$ mES cells, for HaloTag-Eed in wild-type and $Eed^{-/-}$ mES cells, and for HaloTag-Cbx7 in wild-type mES cells. Results are means ± SD. **e** Schematic representation of the nuclear search mechanisms of PcG proteins. The target search time ($\tau_{search}$, solid curved arrowhead) denotes PcG proteins that find a specific site. We assume that the nucleosomal characteristics of the long-lived and short-lived PcG-binding sites are different, as depicted by different size and color. Magenta cylinder represents nucleosomes to which PcG proteins transiently bind, while red cylinder indicates nucleosomes to which PcG proteins specifically bind. **f** Search time ($\tau_{search}$) of the long-lived population for HaloTag-Ezh2 in wild-type and $Ezh2^{-/-}$ mES cells, for HaloTag-Eed in wild-type and $Eed^{-/-}$ mES cells, and for HaloTag-Cbx7 in wild-type mES cells. Results are means ± SD from three biological replicates

bound molecules. We fitted the survival curves by a double or single exponential decay function to generate the short-lived ($f_{1tb}$ and $\tau_{tb}$) and long-lived ($f_{1sb}$ and $\tau_{sb}$) population (Fig. 2b and Methods). After correcting for photobleaching, we estimated the $\tau_{tb}$ of HaloTag-Ezh2 and HaloTag-Eed as ~0.9 s and their $\tau_{sb}$ as ~10 s in wild-type mES cells (Fig. 2c and Supplementary Table 2). The residence times of HaloTag-Ezh2 and HaloTag-Eed in their corresponding knockout mES cells were similar to that in wild-type mES cells (Fig. 2c and Supplementary Table 2). The residence time of PRC2 is similar to that of some transcription factors reported so far[33–39]. Since the control HaloTag-NLS exhibited a negligible long-lived population (Fig. 2c and Supplementary Table 2), we refer to the long-lived molecules as ones that bind stably to specific sites and the short-lived molecules as ones that bind transiently to non-specific sites. It should be noted that we assume that the specific sites have similar affinity for PcG proteins.

Since the level of Cbx7 bound to chromatin is two-fold larger than that of Ezh2 and Eed, we compared their residence times. Interestingly, our results indicated that the residence times of both short-lived and long-lived HaloTag-Cbx7 are nearly identical to that of Ezh2 and Eed (Fig. 2c and Supplementary Table 2), suggesting that PRC2 and Cbx7 exhibit similar stability on chromatin.

The bound level ($F_1$) of PcG proteins is the sum of the short-lived ($F_{1tb}$) and long-lived ($F_{1sb}$) fractions (Methods). We estimated that within wild-type and $Ezh2^{-/-}$ mES cells, ~18% ($F_{1tb}$) of HaloTag-Ezh2 molecules bind transiently to chromatin and ~4.4% ($F_{1sb}$) bind stably to chromatin (Fig. 2d and Supplementary Table 2). Similar analysis of HaloTag-Eed in wild-type and $Eed^{-/-}$ mES cells indicated that ~17% ($F_{1tb}$) binds transiently to chromatin and ~4.2% ($F_{1sb}$) binds stably to chromatin (Fig. 2d and Supplementary Table 2), indicating that Ezh2 and Eed have similar short-lived and long-lived fractions. Live-cell SMT analysis of HaloTag-Cbx7 indicated that ~27% ($F_{1tb}$) of molecules bind transiently to chromatin and ~13% ($F_{1sb}$) bind stably to chromatin.

Taken together, our analyses show that PRC2 and Cbx7 exhibit similar stability (residence time) on chromatin. Thus, their target search processes determine their fractions bound to chromatin.

**PRC2 and Cbx7 exhibit distinct nuclear search dynamics**. To characterize the nuclear search dynamics, we analyzed the target search time ($\tau_{search}$) of PRC2 and Cbx7 (Fig. 2e and Methods). Within mES cells, after dissociating from a specific site, HaloTag-Ezh2 required ~210 s to find another specific site (Fig. 2f and Supplementary Table 2). HaloTag-Eed and HaloTag-Ezh2 had similar times when finding specific sites (Fig. 2f and Supplementary Table 2). The search times of HaloTag-Ezh2 and HaloTag-Eed in their corresponding knockout mES cells were similar to that in wild-type mES cells (Fig. 2f and Supplementary Table 2). In contrast, HaloTag-Cbx7 took ~64 s to find a specific site (Fig. 2f and Supplementary Table 2). The short search time of HaloTag-Cbx7 is due to the fact that HaloTag-Cbx7 takes fewer trials to find a specific site than PRC2, and spends less time between two binding events ($\tau_{3D}$) (Supplementary Table 2 and Methods). Together, our data indicate that Cbx7 and PRC2 have similar stability on chromatin, but Cbx7 is more efficient at finding specific sites than PRC2. Thus, the long-lived fraction of Cbx7 is larger than that of PRC2.

**Effects of Eed and Suz12 on the dynamics of Ezh2**. The minimal catalytically active PRC2 is comprised of Ezh2, Eed, and Suz12[40]. Eed recognizes H3K27me3, which is critical for the propagation of H3K27me3[41,42]. We determined whether Eed is required for

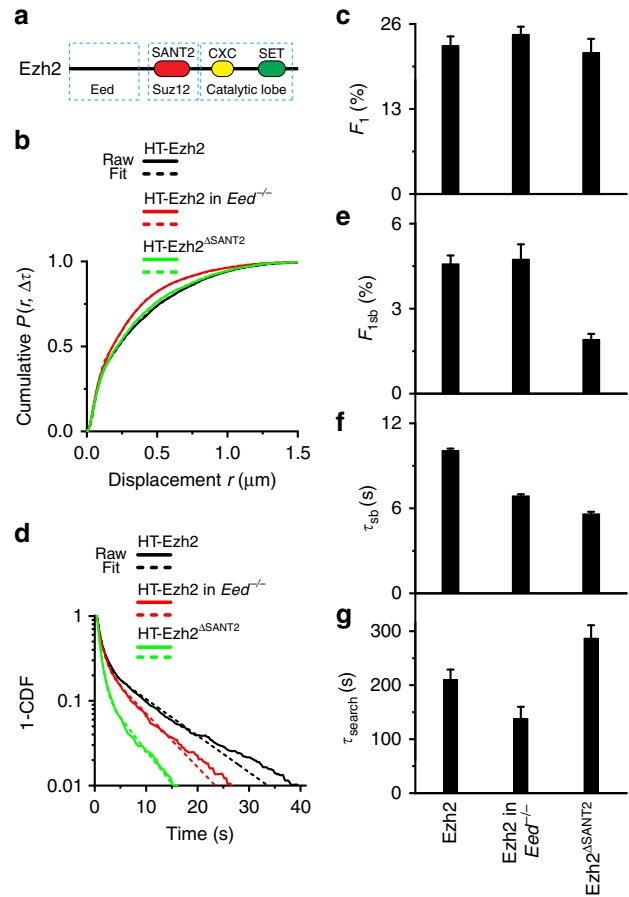

**Fig. 3** Eed and Suz12 are required for the stability of Ezh2 on chromatin, but have different effects on its target search process. **a** Schematic representation of Ezh2. The N-terminus of Ezh2 interacts with Eed. The SANT2 domain (red rectangle) interacts with Suz12. The C-terminus of Ezh2 is the catalytic lobe, including cysteine-rich domain (CXC, yellow rectangle) and SET domain (green rectangle) that is the catalytic domain of Ezh2. **b** Cumulative distribution of displacements for HaloTag-Ezh2 replicated from Fig. 1, for HaloTag-Ezh2 ($N = 64$ cells, $n = 11437$ displacements) in $Eed^{-/-}$ mES cells, and for HaloTag-Ezh2$^{\Delta SANT2}$ ($N = 71$ cells, $n = 5981$ displacements) in wild-type mES cells. The cumulative distributions were fitted with three populations. Fitted parameters are shown in Supplementary Table 3. **c** Fraction of the chromatin-bound population ($F_1$) obtained from Fig. 3b. Results are means ± SD. **d** Survival probability distribution of the dwell times for HaloTag-Ezh2 replicated from Fig. 2, for HaloTag-Ezh2 ($N = 93$ cells, $n = 2339$ trajectories) in $Eed^{-/-}$ mES cells, and for HaloTag-Ezh2$^{\Delta SANT2}$ ($N = 85$ cells, $n = 2728$ trajectories) in wild-type mES cells. The distributions were fitted with a two-component exponential decay model. Fitted parameters are shown in Supplementary Table 4. **e–g** Fraction (**e**), residence time (**f**), and search time (**g**) of the long-lived population ($F_{1sb}$) for HaloTag-Ezh2 replicated from Fig. 2, for HaloTag-Ezh2 in $Eed^{-/-}$ mES cells, and for HaloTag-Ezh2$^{\Delta SANT2}$ in wild-type mES cells. Results are means ± SD from three biological replicates

the stability and search process of Ezh2. To this end, we integrated *HaloTag-Ezh2* into the genome of $Eed^{-/-}$ mES cells. Interestingly, the depletion of *Eed* had no effect on the chromatin-bound fraction ($F_1$) and the long-lived population ($F_{1sb}$) (Fig. 3c, e; Supplementary Fig. 1, and Supplementary Table 3 and 4), but the residence time of long-lived HaloTag-Ezh2 molecules was reduced by ~30% (from ~10 to ~7.0 s) (Fig. 3d, f, and Supplementary Table 4), suggesting Eed is required for the stability of Ezh2 on chromatin. The search time

of HaloTag-Ezh2 in $Eed^{-/-}$ mES cells was ~70% of that in wild-type mES cells (Fig. 3g and Supplementary Table 4). Thus, our analyses demonstrate that the depletion of $Eed$ destabilizes Ezh2 on chromatin and shortens the search process of Ezh2, thereby having no effect on its long-lived fraction.

Suz12 is essential for the PRC2 activity[40,43]. To understand how Suz12 contributes to the binding kinetics and search process of Ezh2, we deleted the SANT2 domain of Ezh2 ($Ezh2^{\Delta SANT2}$) since the SANT2 domain interacts with Suz12[24] (Fig. 3a). The bound fraction ($F_1$) of HaloTag-$Ezh2^{\Delta SANT2}$ was similar to that of HaloTag-Ezh2 (Fig. 3b, c; Supplementary Fig. 1, and Supplementary Table 3); however, the long-lived fraction was reduced by over 50% (Fig. 3e and Supplementary Table 4). The deletion of the SANT2 domain also resulted in a ~50% reduction in the residence time of long-lived molecules (Fig. 3d, f; Supplementary Table 4). The search time of HaloTag-$Ezh2^{\Delta SANT2}$ increased (Fig. 3g and Supplementary Table 4). Thus, our results indicate that the disruption of interactions between Ezh2 and Suz12 destabilizes Ezh2 on chromatin and prolongs its search process, thereby reducing its long-lived fraction.

**Effects of H3.3K27M on the bound level of Cbx7 and Ezh2.** Since H3.3K27 is the substrate of Ezh2 and the Ezh2-catalyzed product H3K27me3 is critical for the targeting of Cbx7 to chromatin[3–6,28], we investigated the effects of the H3.3K27M mutation on the kinetic fractions of Ezh2 and Cbx7. To this end, we established HEK293T cell lines that stably express pairs of *HaloTag-Ezh2/H3.3-FLAG*, *HaloTag-Ezh2/H3.3K27M-FLAG*, *HaloTag-Cbx7/H3.3-FLAG*, or *HaloTag-Cbx7/H3.3K27M-FLAG*. We showed that the expression of *H3.3K27M* results in a ~7.0-fold reduction of the H3K27me3 level compared to the expression of *H3.3* (Fig. 4a and Supplementary Fig. 2). This is consistent with previous reports[20,22]. Live-cell SMT analysis demonstrated that the fraction ($F_1$) of HaloTag-Ezh2 bound to chromatin in *H3.3*-expressing cells is the same as that in *H3.3K27M*-expressing cells (Fig. 4b, c; Supplementary Fig. 3, and Supplementary Table 5), suggesting that H3.3K27M does not influence the chromatin-bound level of Ezh2. The diffusion coefficients of Ezh2 in *H3.3*-expressing and *H3.3K27M*-expressing cells were similar, consistent with previous results measured from FRAP and fluorescence correlation microscopy (FCS)[44]. In contrast, the expression of *H3.3K27M* resulted in a ~30% reduction in the level of HaloTag-Cbx7 bound to chromatin in comparison with the expression of *H3.3* (Fig. 4b, c; Supplementary Fig. 3, and Supplementary Table 5), indicating that H3.3K27M alters the bound level of Cbx7 with chromatin. These results are consistent with the targeting of Cbx7 to chromatin depending on H3K27me3[28]. Since motion blur of molecules can potentially lead to an overestimation of chromatin-bound fraction, we performed SMT at a series of exposure times (5, 10, and 20 ms) with a consistent laser power (Supplementary Fig. 4). Motion blur led to a slight overestimation of PcG proteins bound to chromatin, but did not affect the conclusion. Thus, H3.3K27M has negligible effects on the level of Ezh2 bound to chromatin, but affects the level of Cbx7 bound to chromatin instead (Supplementary Fig. 4 and Supplementary Table 5).

To test the effects of the H3.3K27M mutation on the bound fractions of Ezh2 and Cbx7 in more physiologically relevant conditions, we stably expressed *HaloTag-Ezh2* and *HaloTag-Cbx7* in a DIPG patient-derived tumor cell line (SF8628) containing the heterozygous H3.3K27M mutation and in a control brain tumor cell line (9427)[22]. Our analysis indicated that the H3K27me3 level in SF8628 is reduced by ~2.0-fold in comparison with 9427 cells (Fig. 4d; Supplementary Fig. 2). Similar live-cell SMT analysis demonstrated that the bound fraction of HaloTag-Ezh2 in SF8628

cells is almost identical to that in 9427 cells; however, the bound level of HaloTag-Cbx7 in SF8628 cells is reduced by ~30% compared to 9427 cells (Fig. 4e, f; Supplementary Fig. 3, and Supplementary Table 5). Analysis under the conditions of different exposure times reached the same conclusion. Together, H3.3K27M has no effect on the level of Ezh2 bound to chromatin; however, there is a reduction in the bound level of Cbx7 (Supplementary Fig. 5 and Supplementary Table 5).

**Effects of H3.3K27M on the binding of Cbx7 and Ezh2.** We determined how H3.3K27M influences the stability of Ezh2 and Cbx7 on chromatin. We found that the residence time ($\tau_{sb} = 15.3$ s) of long-lived HaloTag-Ezh2 molecules is ~1.5-fold longer in HEK293T cells expressing *H3.3K27M* than that ($\tau_{sb} = 10.2$ s) in HEK293T cells expressing *H3.3K27* (Fig. 5a, c, and Supplementary Table 6). Similarly, the residence time ($\tau_{sb} = 11.8$ s) of long-lived Ezh2 molecules in patient-derived SF8628 cells was ~1.4-fold longer than that in control 9427 cells ($\tau_{sb} = 8.7$ s) (Fig. 5b, c and Supplementary Table 6). The fraction of long-lived HaloTag-Ezh2 in cells with *H3.3K27M* mutant was similar to that in cells with wild-type *H3.3* (Fig. 5d and Supplementary Table 6). These data suggest that H3.3K27M stabilizes Ezh2 on chromatin, but has no effect on its long-lived fraction.

In contrast to Ezh2, we found that the residence time of long-lived HaloTag-Cbx7 molecules in cells carrying *H3.3K27M* is similar to that in cells carrying *H3.3* (Fig. 5a–c, and Supplementary Table 6). However, the fraction of long-lived HaloTag-Cbx7 in cells expressing *H3.3* was ~1.5-fold larger than that in cells expressing *H3.3K27M* (Fig. 5d and Supplementary Table 6). These data indicate that H3.3K27M has no effect on the stability of Cbx7 on chromatin, but reduces its fraction bound to chromatin.

**H3.3K27M prolongs the search processes of Ezh2 and Cbx7.** Next, we determined how H3.3K27M affects the search processes of Ezh2 and Cbx7. We found that HaloTag-Ezh2 in cells with *H3.3K27M* takes ~1.4-fold longer time to find specific sites than in cells with *H3.3* (Fig. 5e and Supplementary Table 6). Our analyses showed that HaloTag-Cbx7 in cells carrying *H3.3K27M* takes ~1.5-fold longer time to find specific sites than in cells carrying *H3.3* (Fig. 5e and Supplementary Table 6). These data indicate that H3.3K27M prolongs the search processes of Ezh2 and Cbx7.

In summary, H3.3K27M stabilizes Ezh2 on chromatin and prolongs its search process, thereby having no effect on its level bound to chromatin (Fig. 5f). H3.3K27M has no effect on the stability of Cbx7 on chromatin, but lengthens its search time, thereby reducing its fraction on chromatin (Fig. 5f).

**H3.3K27M stabilizes PRC2 on nucleosomes.** Since PRC2 functions as a multiprotein complex, it is possible that other subunits rather than Ezh2 bind H3.3K27M. Since PRC2 recognizes H3.3K27M via its catalytic lobe in vitro[24,45], we tested whether H3.3K27M stabilizes the catalytic lobe on chromatin in vivo. We fused the catalytic lobe with HaloTag to generate HaloTag-$Ezh2^{Catalytic}$ (Fig. 3a). We established HEK293T cell lines that stably express *HaloTag-Ezh2^{Catalytic}/H3.3-FLAG* or *HaloTag-Ezh2^{Catalytic}/H3.3K27M-FLAG*. Similar to HaloTag-Ezh2, the fraction of HaloTag-$Ezh2^{Catalytic}$ bound to chromatin in HEK293T cells with *H3.3K27M-FLAG* was similar to that in HEK293T cells with *H3.3-FLAG* (Fig. 6a, b, d; Supplementary Fig. 6, and Supplementary Tables 7 and 8). The residence time (~7.3 s) and search time (~277 s) of HaloTag-$Ezh2^{Catalytic}$ in *H3.3K27M-FLAG*-expressing cells were both ~1.6-fold of that in *H3.3-FLAG*-expressing cells (Fig. 6c, e, f, and Supplementary

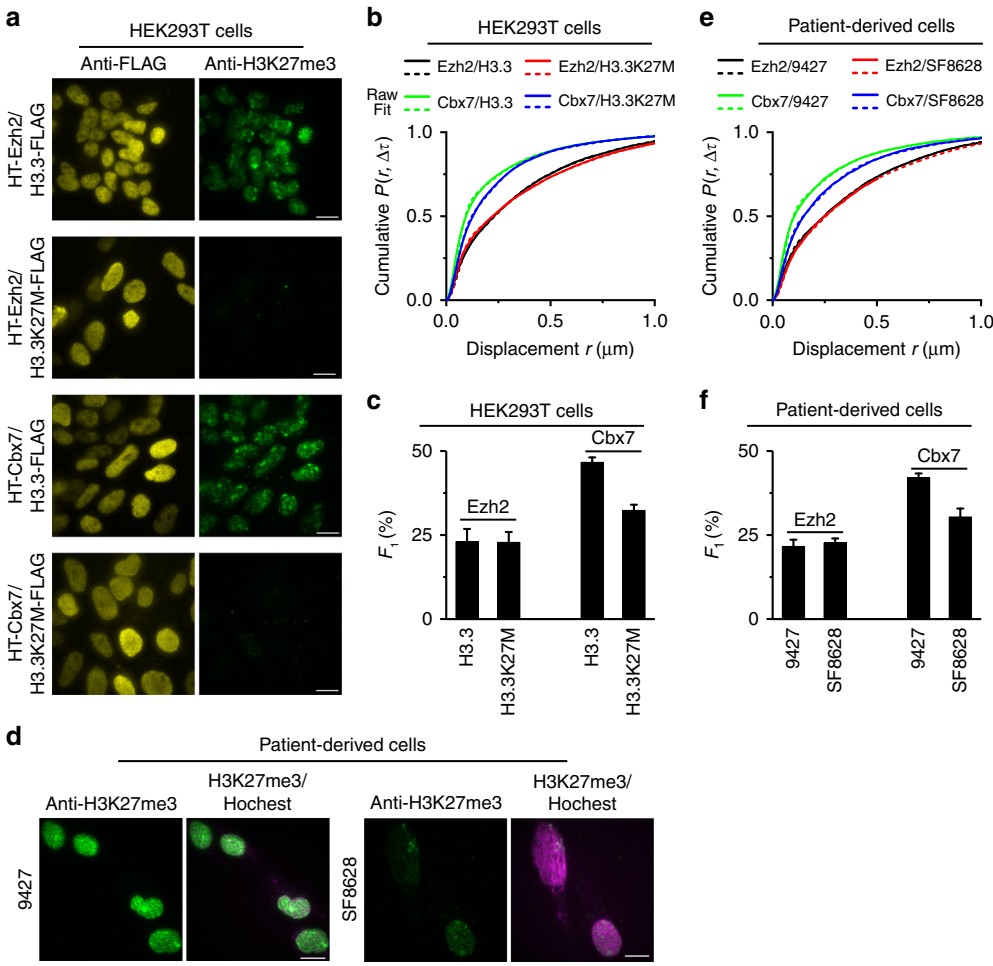

**Fig. 4** The DIPG H3.3K27M mutation reduces the Cbx7 level bound to chromatin, but has no effect on Ezh2. **a** Co-immunostaining of FLAG (yellow) and H3K27me3 (green) in HEK293T cells expressing *HaloTag-Ezh2/H3.3-FLAG*, *HaloTag-Ezh2/H3.3K27M-FLAG*, *HaloTag-Cbx7/H3.3-FLAG*, or *HaloTag-Cbx7/H3.3K27M-FLAG*. Quantification analysis showed that the H3K27me3 level in HEK293T cells expressing *H3.3K27M-FLAG* is ∼7.0-fold lower compared to that expressing *H3.3-FLAG* (Supplementary Fig. 2). Scale bar, 5.0 μm. **b** Cumulative distribution of displacements for HaloTag-Ezh2 in HEK293T cells expressing *H3.3-FLAG* ($N = 54$ cells, $n = 9034$ displacements) or *H3.3K27M-FLAG* ($N = 127$ cells, $n = 22,273$ displacements), and for HaloTag-Cbx7 in HEK293T cells expressing *H3.3-FLAG* ($N = 78$ cells, $n = 14,101$ displacements) or *H3.3K27M-FLAG* ($N = 89$ cells, $n = 15,755$ displacements). The distributions were decomposed into three populations. Fitted parameters are shown in Supplementary Table 5. **c** Fraction of the chromatin-bound population ($F_1$) for HaloTag-Ezh2 or HaloTag-Cbx7 in HEK293T cells expressing *H3.3-FLAG* or *H3.3K27M-FLAG*. The data were obtained from Fig. 4b. Results are means ± SD. **d** Immunostaining of H3K27me3 in patient-derived SF8628 and control brain tumor 9427 cells by using antibody directed against H3K27me3 (green). DNA was stained with hoechst (red). Overlay images are shown. The H3K27me3 level in 9427 cells was ∼2.0-fold higher than that in SF8628 cells (Supplementary Fig. 2). Scale bar, 5.0 μm. **e** Cumulative distribution of displacements for HaloTag-Ezh2 in 9427 ($N = 73$ cells, $n = 6806$ displacements) and SF8628 ($N = 43$ cells, $n = 6900$ displacements) cells, and for HaloTag-Cbx7 in 9427 ($N = 37$ cells, $n = 5280$ displacements) and SF8628 ($N = 32$ cells, $n = 8181$ displacements) cells. The histograms were fitted with three components. Fitted data are shown in Supplementary Table 5. **f** Fraction of the chromatin-bound population ($F_1$) for HaloTag-Ezh2 in 9427 and SF8628 cells and for HaloTag-Cbx7 in 9427 and SF8628 cells. The data were obtained from **e**. Results are means ± SD from three biological replicates

Table 8). These data support that H3.3K27M stabilizes PRC2 on chromatin *via* its interactions with Ezh2 in vivo.

Since cells have nucleosomes with distinct properties, it is possible that other nucleosomes rather than H3.3K27M-nucleosome influence the PRC2 binding. To this end, we carried out in vitro single-molecule binding assays (Fig. 6g and Supplementary Fig. 7). We reconstituted mononucleosomes labeled with biotin and Alexa Fluor 488 at each end of the nucleosomal DNA. The labeled nucleosomes were tethered to a quartz coverslip of a flow cell that had been passivated and functionalized with NeutrAvidin. We integrated *FLAG-HaloTag-Ezh2* into the genome of *Ezh2*$^{-/-}$ mES cells. JF$_{646}$-labeled Ezh2-PRC2 was enriched by anti-FLAG antibody and then the eluent was loaded into the flow cell. We recorded the positions of individual nucleosomes and followed the duration of JF$_{646}$-labeled Ezh2-PRC2 staying on nucleosomes. The dwell time of individual JF$_{646}$-labeled Ezh2-PRC2 molecules were directly measured as the lifetime of the fluorescence spots. The cumulative frequency distribution of dwell times was fitted with a one-component exponential decay function (Fig. 6h). We found that JF$_{646}$-labeled Ezh2-PRC2 binds to H3.3-nucleosome with the residence time ∼257 s and to H3.3K27M-nucleosome with the residence time ∼404 s (Fig. 6i). These data indicate that H3.3K27M-nucleosomes have better affinity for Ezh2 than H3.3-nucleosomes, which is consistent with the in vivo results.

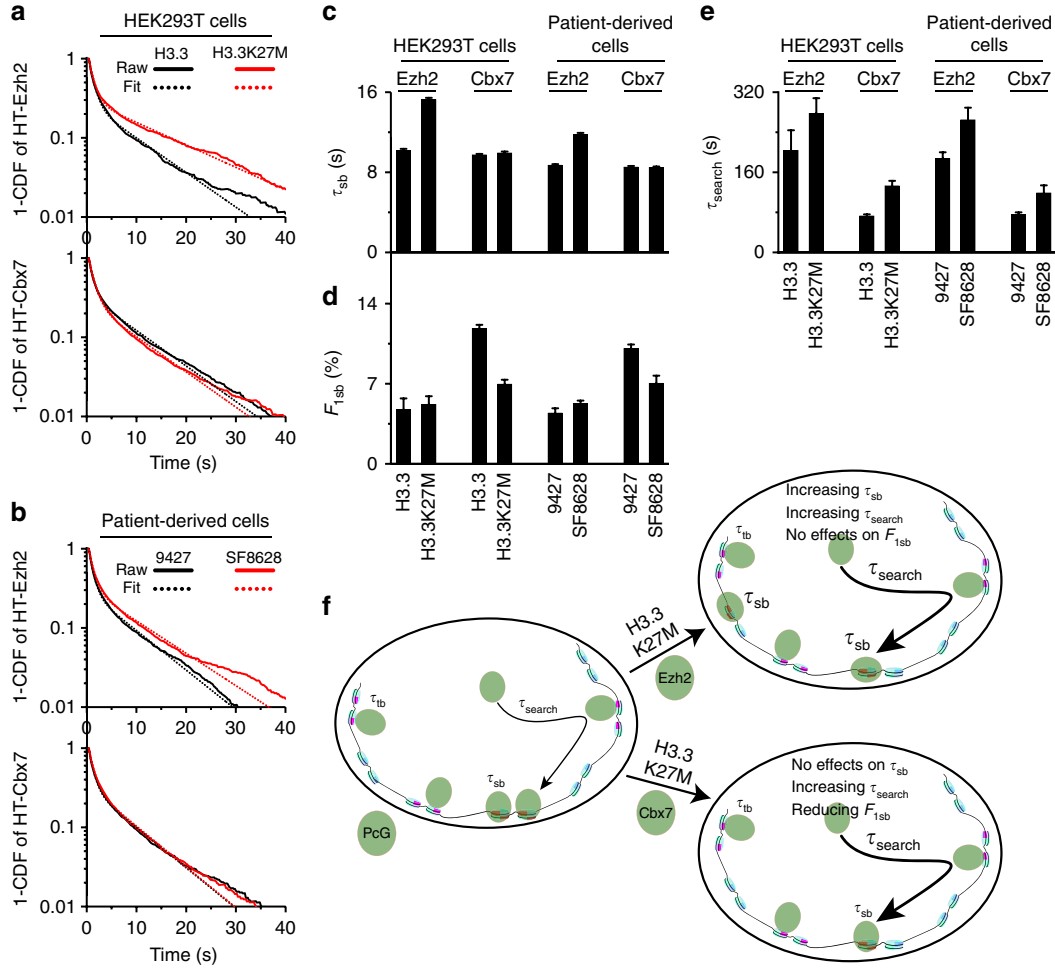

**Fig. 5** The DIPG H3.3K27M mutation prolongs the search processes of Ezh2 and Cbx7, but has different effects on their stability on chromatin. **a** Survival probability distribution of the dwell times for HaloTag-Ezh2 in HEK293T cells expressing *H3.3-FLAG* ($N = 211$ cells, $n = 3825$ trajectories) or *H3.3K27M-FLAG* ($N = 160$ cells, $n = 3137$ trajectories), and for HaloTag-Cbx7 in HEK293T cells expressing *H3.3-FLAG* ($N = 89$ cells, $n = 8024$ trajectories) or *H3.3K27M-FLAG* ($N = 45$ cells, $n = 6866$ trajectories). The distributions were fitted with a two-component exponential decay model. Fitted data are shown in Supplementary Table 6. **b** Survival probability distribution of the dwell times for HaloTag-Ezh2 in 9427 ($N = 90$ cells, $n = 6078$ trajectories) and SF8628 ($N = 54$ cells, $n = 4485$ trajectories) cells, and for HaloTag-Cbx7 in 9427 ($N = 110$ cells, $n = 4641$ trajectories) and SF8628 ($N = 67$ cells, $n = 5376$ trajectories) cells. The distributions were fitted with a two-component exponential decay model. Fitted parameters are shown in Supplementary Table 6. **c–e** Residence time (**c**), fraction (**d**), and search time (**e**) of the long-lived population ($F_{1sb}$) for HaloTag-Ezh2 and HaloTag-Cbx7. Results are means ± SD from three biological replicates. **f** A proposed model for the effects of H3.3K27M on the binding and search mechanisms of PcG proteins. The H3.3K27M mutation prolongs the residence time and search time of Ezh2, but has no effect on the bound level of Ezh2 (right, top panel). Given that the target search time of Ezh2 is ~20-fold longer than its residence time, we propose that the reduced sampling frequency of Ezh2 for the target sites by prolonging its target search time reprograms the genomic distributions of H3K27me3 and PRC2 (right, top panel). The H3.3K27M mutation has no effect on the residence time of Cbx7 on chromatin; however, increases the target search time of Cbx7, indicating that the prolonged search time by H3K27M reduces the bound level of Cbx7 (right, bottom panel). The thicker arrowhead represents a longer search time and the bigger font depicts a prolonged residence time

**Increasing Cbx7 inhibits the proliferation of DIPG cells**. It is well known that the activities of many nuclear factors are regulated by modulating their concentration in the nucleus. To this end, we varied the expression of *HaloTag-Cbx7* in 9427 and SF8628 cells by changing doxycycline concentrations (Fig. 7a). Immunofluorescence indicated that the protein level of Cbx7 increases with increasing doxycycline concentration (Fig. 7b). SF8628 cells showed reduced proliferation in response to elevating HaloTag-Cbx7 concentration (Fig. 7c, d). In contrast, the proliferation of control 9427 cells was not affected by increasing *HaloTag-Cbx7* concentration (Fig. 7c, d). Doxycycline had no effect on the proliferation of SF8628 and 9427 cells (Fig. 7c, d). These results demonstrate that human DIPG cells grow slowly in response to increasing Cbx7 expression.

**Elevating Cbx7 increases the stability of Cbx7 on chromatin**. The classical kinetic model shows that for a given system, the residence time and search time are set. In living cells, elevating nuclear factor concentrations could potentially alter the characteristics of both chromatin and nuclear factors, thereby changing their stability on chromatin and search process. To understand how elevating Cbx7 concentrations inhibits the proliferation of DIPG cells, we investigated the binding and search kinetics of Cbx7 by increasing its concentration in SF8628 cells. Our analysis found that the total number $\left(F_1^N\right)$ of Cbx7 molecules on chromatin increases upon increasing Cbx7 concentrations (Fig. 8a, b; Supplementary Fig. 8, and Supplementary Table 9). However, the number $\left(F_{1sb}^N\right)$ of long-lived Cbx7 molecules at specific sites remained constant (Fig. 8d and Supplementary

Table 10). Increasing Cbx7 concentrations increases the residence time of long-lived Cbx7 molecules (Fig. 8c, e, and Supplementary Table 10). The search time of Cbx7 also increased when its concentrations were elevated (Fig. 8f and Supplementary

Table 10), which is primarily due to the fact that Cbx7 takes more trials of non-specific chromatin collisions to reach a specific site and increase its $\tau_{3D}$ (Supplementary Table 10). Taken together, our data indicate that elevating Cbx7 levels increases the number

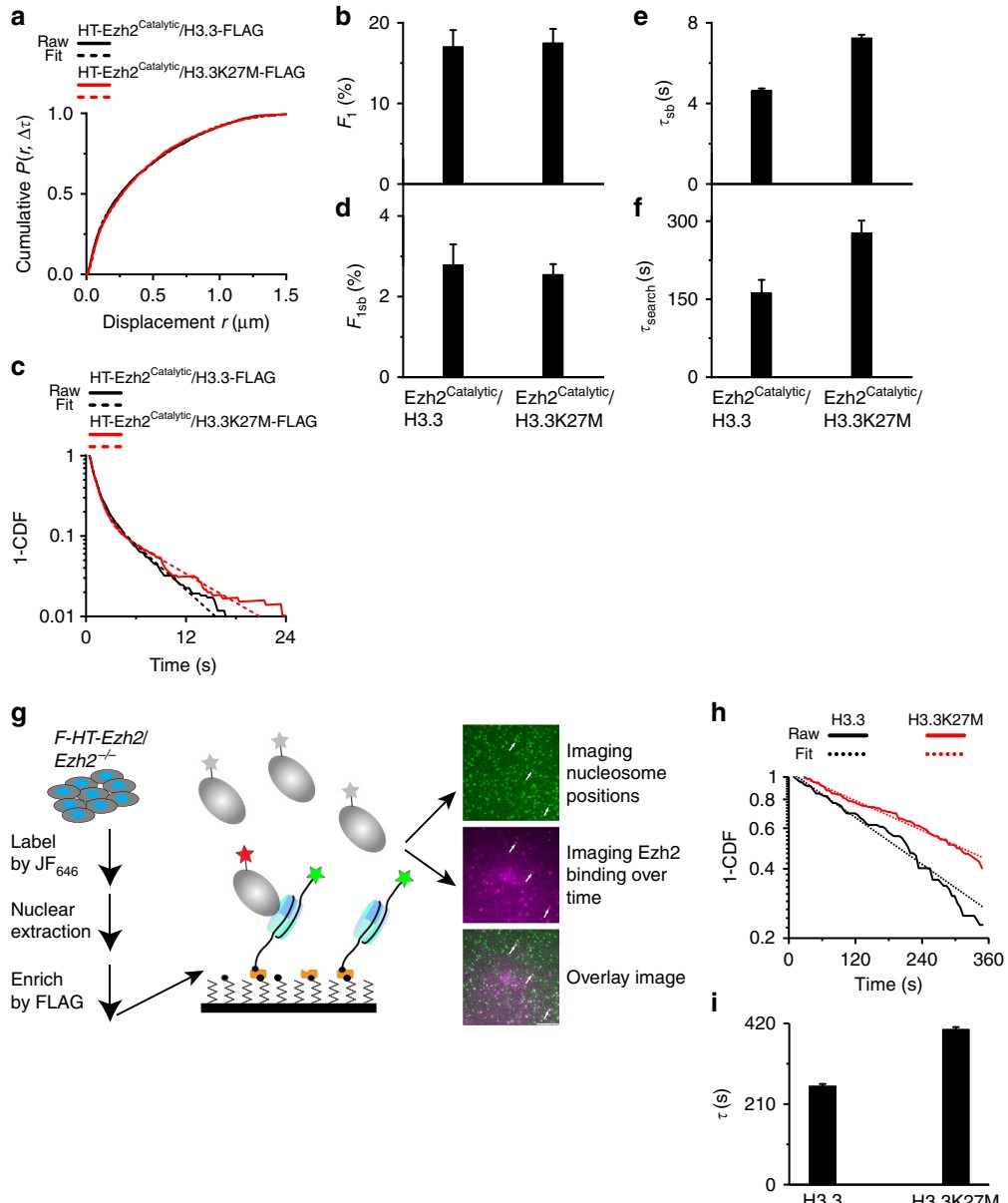

**Fig. 6** The interactions between H3.3K27M and Ezh2 stabilize Ezh2 on chromatin. **a** Cumulative distribution of displacements for HaloTag-Ezh2$^{Catalytic}$ in HEK293T cells expressing *H3.3-FLAG* ($N = 42$ cells, $n = 2365$ displacements) or *H3.3K27M-FLAG* ($N = 36$ cells, $n = 4120$ displacements). The cumulative distributions were fitted with three populations. Ezh2$^{Catalytic}$ contains only the catalytic lobe (Fig. 3a). Fitted parameters are shown in Supplementary Table 7. **b** Fraction of the chromatin-bound population ($F_1$) obtained from **a**. Results are means ± SD. **c** Survival probability distribution of the dwell times for HaloTag-Ezh2$^{Catalytic}$ in HEK293T cells expressing *H3.3-FLAG* ($N = 56$ cells, $n = 1412$ displacements) or *H3.3K27M-FLAG* ($N = 46$ cells, $n = 1042$ displacements). The distributions were fitted with a two-component exponential decay model. Fitted parameters are shown in Supplementary Table 8. **d–f** Fraction (**d**), residence time (**e**), and search time (**f**) of the long-lived population ($F_{1sb}$) for HaloTag-Ezh2$^{Catalytic}$ in HEK293T cells expressing *H3.3-FLAG* or *H3.3K27M-FLAG*. Results are means ± SD. **g** In vitro single-molecule binding assay. FLAG-HaloTag-Ezh2 (*F-HT-Ezh2*) was expressed stably in *Ezh2$^{-/-}$* mES cells and labeled by JF$_{646}$ dyes. FLAG-HaloTag-Ezh2-PRC2 was one-step enriched by FLAG antibody and loaded into the flow cells. Nucleosome labeled with Alexa Fluor®488 was tethered to the surface of a coverslip that had been passivated and functionalized with NeutrAvidin. The position of individual nucleosomes was recorded (green image). We followed the duration of FLAG-HaloTag-Ezh2-PRC2 on nucleosome (magenta image). The colocalization of nucleosome and FLAG-HaloTag-Ezh2 indicates a binding event (overlay image). Arrowheads indicate binding events. Scale bar, 5.0 μm. **h** Survival probability distribution of the dwell times for FLAG-HaloTag-Ezh2-PRC2 on H3.3-nucleosome ($n = 99$ trajectories) and on H3.3K27M-nucleosome ($n = 115$ trajectories). The distributions were fitted with a one-component exponential decay model. **i** Residence time of FLAG-HaloTag-Ezh2-PRC2 on H3.3-nucleosome and on H3.3K27M-nucleosome. Results are means ± SD from three biological replicates

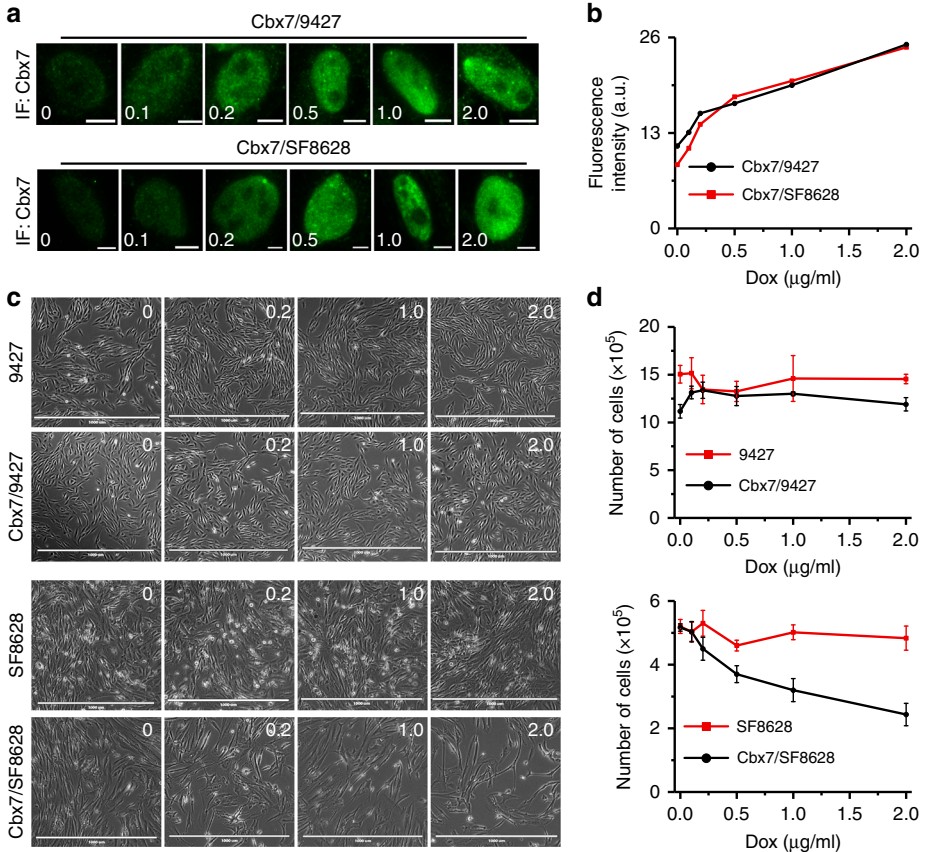

**Fig. 7** Increasing Cbx7 levels inhibits the proliferation of patient-derived DIPG cells. **a** Immunostaining of Cbx7 in 9427 and SF8628 cells carrying transgenic *Cbx7* under the control of TRE-tight promoter. The expression of *Cbx7* was induced by a variety of doxycycline concentrations. Scale bar, 2.0 μm. **b** Quantification of fluorescence intensity of Cbx7 from **a**. **c** Representative images of proliferation response of 9427 and SF8628 cells to increasing levels of Cbx7. Images of cells treated with 0.1 and 0.5 μg per ml of Dox are not shown. Scale bar, 1000 μm. **d** Quantification of proliferation response of 9427 and SF8628 cells to increasing levels of Cbx7. Values shown are the average from triplicate samples for each incubation condition

of short-lived Cbx7 molecules on the non-specific sites and stabilizes Cbx7 on the specific sites; however, has no effect on the number of long-lived Cbx7 molecules on the specific sites.

## Discussion

We have determined the binding and target search kinetics of PRC2 and Cbx7 within living mES cells and elucidated how H3.3K27M alters their binding and target search mechanisms. In the following, we discuss how the prolonged residence time contributes to the inhibition of trimethylation of H3K27 by PRC2 in vitro and how the lengthened target search time regulates the reprogramming of H3K27me3 in vivo. We also discuss how increasing Cbx7 levels inhibits the proliferation of DIPG cells and influences its binding and search mechanisms.

Our live-cell SMT experiments reveal that both PRC2 and Cbx7 scan the nuclear environment by a combination of free diffusion and transient and stable interactions with chromatin, similar to previous reports for transcription factors[36,38,46]. PRC2 and Cbx7 have identical residence times (~10 s), similar to transcription factors[33–38], but employ different target search mechanisms. First, PRC2 needs ~200 s to find a specific site, while Cbx7 takes ~60 s. Second, PRC2 spends ~40 s in free diffusion, but Cbx7 spends ~20 s. Third, the long-lived fraction of Cbx7 is ~2.5-fold larger than PRC2. These data suggest that Cbx7 is more efficient in searching for its targets compared to PRC2. Our analyses highlight that the target search time of PRC2 and Cbx7 determines the difference in their genomic occupancy levels.

H3K27M results in a global reduction in the H3K27me3 level, but selectively retains a subset of loci with H3K27me3 and PRC2[22,23]. Our analysis indicates that H3.3K27M increases the $\tau_{sb}$ of Ezh2 from ~10 to ~15 s and the $\tau_{search}$ of Ezh2 from ~200 to ~280 s. How does this explain the altered genomic occupancy of H3K27me3 and PRC2 by H3K27M? The interval (s) of sampling of specific sites by PRC2 can be described as Eq. 1[38].

$$\frac{(\tau_{sb} + \tau_{search}) \times \text{The number of specific sites}}{\text{The number of PRC2 molecules in the nucleus}} \quad (1)$$

Our single-molecule data provide evidence that H3.3K27M prolongs not only the residence time of PRC2, but also its search time. The search time of Ezh2 is ~20-fold longer than its residence time. A previous report showed that the PRC2 level in *H3.3K27M*-expressing cells is the same as that in *H3.3*-expressing cells[22]. If we assume that the number of specific sites in *H3.3*- and *H3.3K27M*-expressing cells remains unchanged, the search time of PRC2 rather than its residence time contributes greatly to its sampling frequency. Therefore, the lengthened search time increases the interval of sampling of specific PcG-targeted sites. We hypothesize that the reduced sampling frequency has differential effects on Polycomb targets: less effects on strong Polycomb targets (more binding sites), but greater effects on weaker Polycomb targets (less binding sites). This model agrees with a recent study that demonstrated strong Polycomb targets have clusters or long CpG islands (more binding sites) and high PRC2/H3K27me3 enrichment, while weak Polycomb target sites have isolated or short CpG islands (less binding sites) and low PRC2/

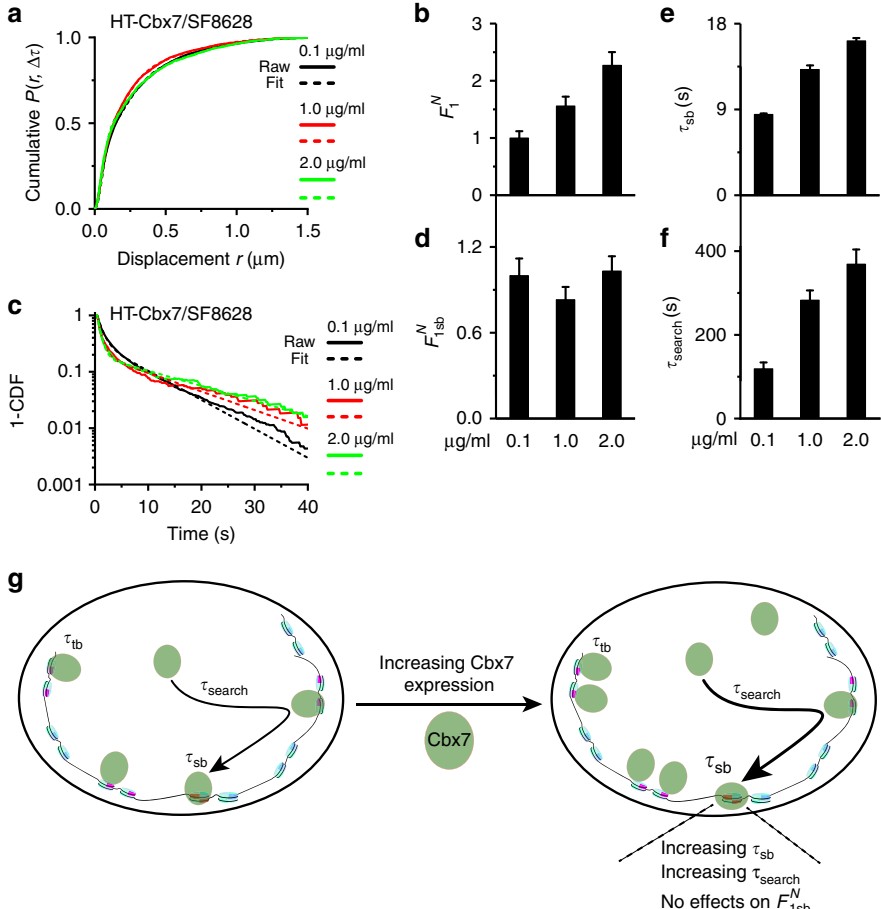

**Fig. 8** Elevating Cbx7 levels increases its stability on chromatin. **a** Cumulative distribution of displacements for HaloTag-Cbx7 in SF8628 cells administrated with 0.1 μg per ml doxycycline replicated from Fig. 4, 1.0 μg per ml ($N = 47$ cells, $n = 7102$ displacements), and 2.0 μg per ml ($N = 38$ cells, $n = 3543$ displacements) doxycycline. The cumulative distributions were fitted with three populations. Fitted parameters are shown in Supplementary Table 9. **b** Normalized chromatin-bound population ($F_1^N$). Unless otherwise indicated, the superscript $N$ represents that a given value was normalized by the Cbx7 level obtained from the immunostaining of Cbx7 (Fig. 7b) and by the chromatin-bound level at 0.1 μg per ml doxycycline. Results are means ± SD. **c** Survival probability distribution of the dwell times for HaloTag-Cbx7 in SF8628 cells administrated with 0.1 μg per ml doxycycline replicated from Fig. 5, 1.0 μg per ml ($N = 49$ cells, $n = 725$ trajectories), and 2.0 μg per ml ($N = 64$ cells, $n = 907$ trajectories) doxycycline. The distributions were fitted with a two-component exponential decay model. Fitted parameters are shown in Supplementary Table 10. **d–f** Normalized long-lived fraction ($F_{1sb}^N$) (**d**), residence time (**e**), and search time (**f**) of the long-lived HaloTag-Cbx7 population. Results are means ± SD from three biological replicates. **g** A proposed mechanism that underpins how increasing Cbx7 levels influences the binding and target search mechanism. Our data indicate that increasing Cbx7 levels has no effect on the number of Cbx7 molecules on the specific sites, however, increases its level bound to the non-specific sites. The residence time of Cbx7 molecules on the specific sites increases upon increasing Cbx7 levels. Increasing Cbx7 levels increases its search time, which is primarily due to the fact that Cbx7 takes more trials of the non-specific chromatin collisions to reach a specific site. The thicker arrowhead represents a longer search time and the bigger font depicts a prolonged residence time

H3K27me3 enrichment[23]. Weak Polycomb targets tend to lose H3K27me3 and PRC2 while strong Polycomb targets are more likely to retain H3K27me3 and PRC2[23]. Thus, our live-cell single-molecule results provide insights into the mechanism that underpins how the reduced sampling frequency of PRC2 reduces the global H3K27me3 level, but retains H3K27me3 at some loci.

A recent genome-wide study showed that H3K27M localizes to transcriptionally active chromatin regions and gene regulatory elements, and PRC2 is largely excluded from chromatin on sites containing H3K27M[26]. However, another recent locus-specific ChIP-qPCR study indicated that H3K27M is not excluded from chromatin regions containing PRC2[23]. The discrepancy could be attributed to the approaches used. Our analysis indicates that ~5% of Ezh2 molecules bind stably to chromatin in both *H3.3*- and *H3.3K27M*-expressing cells and the residence time of these molecules is ~1.5-fold longer in *H3.3K27M*-expressing cells than

*H3.3*-expressing cells. PRC2 and H3K27me3 are present at thousands of loci in DIPG cells[23,26]; therefore, some of the ~5% of Ezh2 molecules are responsible for the trimethylation of these loci in DIPG cells and others are sequestered at H3K27M-containing sites. The sequestered Ezh2 contributes less to the reprogramming of H3K27me3 in vivo, while the reduced sampling frequency results in the loss/retainment of H3K27me3.

Previous studies have shown that H3K27me3 is greatly reduced in *H3.3K27M*-expressing cells[20–22] and is the recruiting factor for Cbx7-PRC1[28]. Our single-molecule studies show that H3.3K27M reduces the level of Cbx7 on chromatin and prolongs its search time, but has no effect on its residence time. The increase of search time of Cbx7 is due to the fact that the number of H3K27me3 is decreased. Thus, our analyses point out that the prolonged search process by H3.3K27M regulates the genomic occupancy of Cbx7.

The minimal components required for the reconstitution of an active PRC2 complex include Ezh2, Eed, and Suz12[40,47,48]. Our analysis demonstrates that the depletion of *Eed* results in a ~30% reduction in both residence time and search time for Ezh2. Consistently, we find that Eed has no effect on the long-lived fraction of Ezh2. Previously, Suz12 was shown to be essential for PRC2 binding to nucleosome[48]. Our analysis shows that the long-lived fraction of Ezh2$^{\Delta SANT2}$ is ~2.5-fold less than Ezh2. In contrast to the *Eed* knockout, the deletion of SANT2 domain not only prolongs the search process of Ezh2, but also reduces its residence time. Studies have shown that Eed and Suz12 are essential for the methyltransferase activity in vivo and in vitro[40,47,48]. Our observations suggest that the number of Ezh2 molecules on chromatin and its target search time may not be a good predictor for the PRC2 activity in vivo, since *Eed* knockout has no effect on the long-lived fraction of Ezh2, and Suz12 and Eed have opposite effects on the target search process of Ezh2. We propose that the residence time of Ezh2 on chromatin may be a better prediction for the methyltransferase activity of PRC2, which is consistent with a recent report in which a prolonged residence time increases the methyltransferase activity of PRC2[49]. The lengthened residence time gives sufficient time to reorient the enzyme on chromatin, which produces a configuration that favors the catalysis.

Studies have shown that H3K27M-nucleosomes are remarkably potent inhibitors of PRC2 and H3K27M peptides bind to the PRC2 active site with above 20-fold higher affinity than the equivalent peptides[20,24,25]. However, a recent study using nucleosomes as substrates showed that the H3K27M mutation has minor effects on PRC2-nucleosome binding[27]. The discrepancy could be due to the experimental conditions, such as the different substrates used (peptides vs. nucleosomes), the composition of PRC2 complexes, and the presence or absence of S-adenosyl methionine. How do our single-molecule results reconcile with the seemingly controversial results? In the binding experiments[27], the binding affinity was characterized by the dissociation constant ($K_D$), a ratio of off-rate vs. on-rate. Our in vivo and in vitro single-molecule analysis demonstrates that the H3K27M oncogenic mutation prolongs the residence time of Ezh2 on H3K27M-nucleosome. The lengthened residence time results in a reduction in the effective concentration of PRC2 by sequestrating PRC2 on H3K27M-nucleosome, which leads to a reduced reaction rate of trimethylation of H3K27 by PRC2 in vitro.

Our data show that increasing expression of *Cbx7* inhibits the proliferation of DIPG cells. A recent study demonstrated that overexpression of *Cbx7* arrests glioma cells in the $G_0/G_1$ phase[15]. It has been well known that the activities of many nuclear factors are regulated by modulating their concentration in the nucleus. Our single-molecule analysis demonstrates that increasing Cbx7 protein levels has no effect on the number of long-lived Cbx7 molecules on the specific sites, but instead prolongs its residence time. We propose that the prolonged residence time facilitates recruiting other co-repressors by Cbx7, thereby repressing gene transcription. Consistently, emerging evidence suggests that the residence time of transcription factors directly correlates with transcriptional output[36,49–52]. Further experiments are needed to understand the causal relationship between residence time and transcriptional activity.

In summary, our analysis immediately suggests that the residence time/stability directly correlates with the functions of PcG proteins and highlights that the target search time plays a critical role in regulating the genomic occupancy of PcG proteins. Perturbing the dynamic equilibrium of the overall system potentially alters cellular physiologies.

## Methods

**Cell culture**. The $Ezh2^{-/-}$ mES cells (Strain 129)[7] were obtained from Dr. Stuart Orkin (Harvard Medical School, Boston, MA). The PGK12.1 mES cells (a heterozygous for Strain PGK and Strain 129)[53] were obtained from Dr. Neil Brockdorff (University of Oxford, UK). The $Eed^{-/-}$ mES cells (Strain 129)[54] were obtained from Dr. Haruhiko Koseki (RIKEN Center for Integrative Medical Sciences, Japan). The patient-derived primary DIPG cells (SF8628) and the control neural stem cells (9427)[22] were obtained from Dr. Zhiguo Zhang (Columbia University, Irving Cancer Research Center, New York). The HEK293T cells were obtained from Dr. Tom K Kerppola (University of Michigan Medical School, Ann Arbor, Michigan). All cell lines were tested negative for mycoplasma contamination by using DAPI DNA staining. SF8628 and 9427 cells were maintained in DMEM medium (D5796; Sigma-Aldrich) supplemented with 10% FBS (97068-085; VWR), 2 mM glutamine (25030–081; Life Technologies), 100 units per ml penicillin-streptomycin (15140–122; Life Technologies) at 37 °C in 5% $CO_2$. The mES cells were maintained in DMEM (D5796; Sigma-Aldrich) supplemented with 15% FBS (97068-085; VWR), 2 mM glutamine (G7513; Life Technologies), 100 units per ml penicillin-streptomycin (15140-122; Life Technologies), 55 μM β-mercaptoethanol (21985-023; Life Technologies), $10^3$ units per ml leukemia inhibitor factor, and 0.1 mM non-essential amino acids (11140050; Life Technologies) at 37 °C in 5% $CO_2$. The HEK293T cells were maintained in DMEM (D5796; Sigma) supplemented with 10% FBS (97068-085; VWR), 2 mM glutamine (G7513; Life Technologies), and 100 units per ml penicillin-streptomycin (15140-122; Life Technologies) at 37 °C in 5% $CO_2$.

**Plasmids**. The plasmids pTRIPZ (M1)-H2A-HT, pTRIPZ (M1)-HT-NLS, and pTRIPZ (M1)-HT-Cbx7 have been deposited at Addgene (https://www.addgene.org/Xiaojun_Ren/)[28]. Note that we have re-named pTRIPZ (M) as pTRIPZ (M1) in this study. We replaced the puromycin-resistant gene in pTRIPZ (M1) with the G418-resistant gene, generating pTRIPZ (M2). We amplified the *HT-Ezh2*, *HT-Eed*, and *HT-Cbx7* sequences and inserted them into pTRIPZ (M2) vector, respectively, generating pTRIPZ (M2)-HT-Ezh2, pTRIPZ (M2)-HT-Eed, and pTRIPZ (M2)-HT-Cbx7. We replaced the TRE-tight promoter in pTRIPZ (M1)-HT-Cbx7 with the CMV promoter amplified from the plasmid pcDNA3.3 (+), generating pTRIPZ (M1)-HT-Cbx7 (ΔTRE). We amplified the *H3.3-FLAG-HA* and *H3.3K27M-FLAG-HA* sequences from pQCXIP-H3.3-FLAG-HA and pQCXIP-H3.3K27M-FLAG-HA[22] and used them to replace HT-Cbx7 in pTRIPZ (M1)-HT-Cbx7 (ΔTRE), generating pTRIPZ (M1)-H3.3-FLAG-HA (ΔTRE) and pTRIPZ (M1)-H3.3K27M-FLAG-HA (ΔTRE). We amplified the *FLAG-HaloTag-Ezh2* sequence and inserted it to pTRIPZ (M1), generating pTRIPZ (M1)-FLAG-HT-Ezh2. To generate Ezh2 variants fused with HaloTag, the Ezh2 sequence in the plasmid pTRIPZ (M1)-HT-Ezh2 was replaced with the Ezh2 variant sequences. The Ezh2 variants were as follows: 1) Ezh2$^{\Delta SANT2}$, deletion of the SANT2 domain (amino acid 318–478), and 2) Ezh2$^{Catalytic}$, amino acid sequence 485–747. The sequences encoding the fusion genes have been verified by DNA sequencing and the plasmids are available at Addgene (https://www.addgene.org/Xiaojun_Ren/).

**Establishing cell lines**. Establishing the cell lines by lentivirus transduction was carried out by using the published procedure[28,55,56]. HEK293T cells at 85–90% confluency were co-transfected with 21 μg pTRIPZ (M) containing the fusion gene, 21 μg psPAX2, and 10.5 μg pMD2.G by using calcium phosphate precipitation. At the time of 12 h after transfection, the medium was replaced with 10 ml DMEM supplemented with 10% FBS, 2 mM L-glutamine, and 100 units per ml penicillin G sodium. At the time of 50 h after medium change, the medium was harvested to transduce cells in the presence of 8.0 μg per ml polybrene (H9268; Sigma-Aldrich, St Louis, MO). For co-transducing multiple genes, lentiviruses were produced separately and mixed at the time of transduction. At the time of 72 h after transduction, infected cells were selected by using 1.0–2.0 μg per ml of puromycin (P8833; Sigma-Aldrich, St Louis, MO) and/or 600–800 μg per ml G418 disulfate salt (A1720, Sigma-Aldrich, St Louis, MO). Unless otherwise indicated, for live-cell single-molecule imaging experiments, the fusions were expressed at the basal level without administrating doxycycline.

**Immunofluorescence**. Wild-type (PGK12.1), $Eed^{-/-}$, $Ezh2^{-/-}$, HaloTag-Eed/$Eed^{-/-}$, and HaloTag-Ezh2/$Ezh2^{-/-}$ mES cells and SF8628, 9427, HaloTag-Cbx7/SF8628, and HaloTag-Cbx7/9427 cells were fixed using 2.0% paraformaldehyde. After permeabilizing with 0.2% Triton X-100, we washed the cells with basic blocking buffer (10 mM PBS pH 7.2, 0.1% Triton X-100, and 0.05% Tween 20) and then with blocking buffer (the basic blocking buffer plus 3% goat serum and 3% bovine serum albumin) overnight. Anti-H3K27me3 antibody (9733; Cell Signaling Technology; 1:200 dilutions) or anti-Cbx7 antibody (ab21873; Abcam; 1:250 dilutions) was incubated with the cells for 2 h at room temperature. After washing with the basic blocking buffer, we incubated cells with Alexa Fluor® 488-labeled goat anti–rabbit antibody (A-11008; Life Technologies; 1:1000 dilutions) for 1 h. After incubating with 25 ng per ml hoechst, we washed and mounted the cells on slides with ProLong Antifade reagents (P7481; Life Technologies).

**Growth curve**. SF8628, 9427, HaloTag-Cbx7/SF8628, and HaloTag-Cbx7/9427 were seeded evenly into 6-well plates with the density at ~7 × $10^4$ cells per ml and

administrated with a series of doxycycline concentration (0, 0.1, 0.2, 0.5, 1, and 2 μg per ml). We incubated the cells at 37 °C in 5% $CO_2$ for 7 days. The cells were imaged and then counted using hemocytometer. The experiments were repeated three times.

**Live-cell SMT.** Labeling HaloTag fusion protein with JF$_{549}$ in live cells: Labeling HaloTag fusion protein with JF$_{549}$ in living cells was performed using the previous procedure[28]. Cells stably expressing HaloTag fusion protein were seeded to gelation-coated cover-glass dish made in the laboratory. Following 24 h culture, cells were treated using several concentrations (a range of 30–100 pM) of JF$_{549}$ to achieve 5–20 labeled molecules per frame. After 15-min incubation at 37 °C in 5% $CO_2$, cells were washed with cell culture medium and incubated in the cell culture medium for 30 min at 37 °C in 5% $CO_2$. The medium was replaced with the live-cell imaging medium (A1896701, FluoroBrite DMEM, Life Technologies). Cells were maintained at 37 °C using a heater controller (TC-324; Warner Instrument) during imaging. Each dish was imaged for <1.5 h.

Single-molecule optical setup and image acquisition: The optical setup for live-cell SMT is as follows. A Zeiss Axio Observer D1 Manual Microscopy (Zeiss, Germany) equipped with an Alpha Plan-Apochromatic ×100/1.46 NA Oil-immersion Objective (Zeiss, Germany) and an Evolve 512 × 512 EMCCD camera with pixel size 16 μm (Photometrics, Tucson, AZ) was used for single-molecule tracking. The overall magnification was ×250 through additional ×2.5 magnification on the emission pathway. The HILO illumination mode was used to avoid stray-light reflection and reduce background from cell auto-fluorescence[30]. For the excitation and emission of JF$_{549}$, a Brightline® single-band laser filter set (Semrock; excitation filter: FF01-561/14, emission filter: FF01-609/54, and dichroic mirror: Di02-R561-25) was used. The microscope and the EMCCD camera were controlled by Slidebook 6.0 software. Laser power intensity of ~10.6 mW per cm$^2$ and ~3.4 mW per cm$^2$ were used to study diffusion components and residence times, respectively.

Single-molecule localization and tracking: We used U-track algorithm for tracking and linking single particles[57]. U-track parameters used in this study are listed in Supplementary Table 11. Prior to analysis, we visually checked stacks of images. About one-third of stacks were discarded due to movement and drift. A Matlab script was developed to process the output of 2D tracking from the u-track.

Extraction of bound fractions and diffusion constants: We extracted kinetic fractions and diffusion constants of PcG proteins using Spot-On[31], which was developed by Hansen et al.[31,46] based on a kinetic modeling framework described by Mazza[33]. Briefly, for a Brownian motion in two dimensions, the probability that a molecule starting at the origin will be at position $r$ at time $\Delta\tau$ is given as follows (Eq. 2).

$$p(r, \Delta\tau) = \frac{r}{2D\Delta\tau} \exp\left(\frac{-r^2}{4D\Delta\tau}\right) \quad (2)$$

The displacement histograms were fitted with the 3-state model (Eq. 3) by using Matlab.

$$p(r, \Delta\tau) = F_1 \frac{r}{2(D_1\Delta\tau + \sigma^2)} \exp\left(\frac{-r^2}{4(D_1\Delta\tau + \sigma^2)}\right)$$
$$+ Z_{CORR}(\Delta\tau, \Delta Z_{Corr}, D_2) F_2 \frac{r}{2(D_2\Delta\tau + \sigma^2)} \exp\left(\frac{-r^2}{4(D_2\Delta\tau + \sigma^2)}\right) \quad (3)$$
$$+ Z_{CORR}(\Delta\tau, \Delta Z_{Corr}, D_3) F_3 \frac{r}{2(D_3\Delta\tau + \sigma^2)} \exp\left(\frac{-r^2}{4(D_3\Delta\tau + \sigma^2)}\right)$$

where $\sigma$ is the single-molecule localization error. We denoted the $F_1$ component as the chromatin-bound population, $F_2$ as the confined diffusion population, and $F_3$ as the free diffusion population. Please note that $F_2$ can also be molecules that non-specifically associate with chromatin. The sum of $F_1$, $F_2$ and $F_3$ equals 1.0. $Z_{CORR}$ is a correct factor for de-focalization bias that considers the probability of molecules moving out of the axial detection range, $\Delta Z$, during delay time $\Delta\tau = 30$ ms (Eqs. 4 and 5).

$$Z_{CORR}(\Delta\tau, \Delta Z_{corr}, D) = \frac{1}{\Delta Z_{corr}} \int_{\frac{-\Delta Z_{corr}}{2}}^{\frac{\Delta Z_{corr}}{2}} \left\{ 1 - \sum_{n=0}^{\infty} (-1)^n \left[ erfc\left(\frac{\frac{(2n+1)\Delta Z_{corr}}{2} - Z}{\sqrt{4D\Delta\tau}}\right) \right. \right.$$
$$\left. \left. + erfc\left(\frac{\frac{(2n+1)\Delta Z_{corr}}{2} + Z}{\sqrt{4D\Delta\tau}}\right) \right] \right\} dZ \quad (4)$$

$$\Delta Z_{corr}(\Delta Z, \Delta\tau, D) = \Delta Z + a(\Delta Z, \Delta\tau)\sqrt{D} + b(\Delta Z, \Delta\tau) \quad (5)$$

where the coefficients $(a, b)$ are from Monte Carlo simulations for a given diffusion constant and are implemented in Spot-On[31]. The parameters used in the fitting are listed in Supplementary Table 12.

In the above analysis, we assumed that the HaloTag-labeled molecules diffuse isotropically along the three-dimensional axes $X$, $Y$, and $Z$. Thus, the $XY$ projection

data reflect the 3D diffusion of the molecules. Live-cell SMT intends overestimating the chromatin-bound molecules against freely diffusing molecules for two major reasons. One bias is that 3D freely diffusing molecules will jump out of the $Z$ axial detection window[58], which has been corrected by the Spot-On[31]. Another bias is that 3D freely diffusing molecules tend to "motion blur" because they can move several pixels during a short exposure time. The extent of the bias by motion blur is sensitive to the experimental conditions and the localization algorithm[31]. To test whether motion blur influences our conclusions, we performed a series of experiments in which we fixed the lag time (30 ms), but varied the exposure times (5, 10, 20 , and 30 ms) (Supplementary Figs. 4, 5). Motion blur caused a slight overestimation of bound PcG proteins (Supplementary Table 5), but did not affect our conclusions. It should be noted that even at <5 ms exposure time, motion blur still contributes to the overestimation of bound PcG proteins. However, our optic setup limits us to further reduce the exposure time due to signal-to-noise. Finally, we assumed that the state transition between a bound molecule and an unbound molecule with the lag time (30 ms) is negligible. Under our analysis, if we consider the residence time as $\tau_{tb} = 1.0$ s, the probability that a bound PcG unbinds during 30 ms is <3%, which was calculated as follows (Eq. 6).

$$P_{switch} = 1 - e^{\left(-\frac{\Delta\tau}{\tau_{sb}}\right)} \quad (6)$$

Determination of residence time: To determine residence time, we performed 30-ms integration time and 170-ms dark time. Molecules with $D_m \leq 0.032$ μm$^2$ per s were considered as chromatin-bound ones. We counted the dwell time of individual molecules as their track length. We estimated the residence time using the cumulative frequency distribution of dwell times as described in[28,33,34,59]. The cumulative frequency distributions of dwell times were normalized for photobleaching as described in[28,33,34,59]. The normalized cumulative frequency distributions were fitted with a one- or two-component exponential decay function (Eq. 7) based the $F$-test implemented in OriginLab.

$$y = (1 - f_{1sb})e^{-\tau/\tau_{tb}} + f_{1sb}e^{-\tau/\tau_{sb}} \quad (7)$$

where $f_{1sb}$ is the long-lived fraction of PcG on chromatin, and $\tau_{tb}$ and $\tau_{sb}$ are the residence time of the short-lived and long-lived components, respectively. In the following, we will use "tb" and "sb" as abbreviations for the short-lived and long-lived population, respectively. Among the total PcG molecules within cells, the fraction of the short-lived component ($F_{1tb}$) and the long-lived component ($F_{1sb}$) were calculated as follows (Eqs. 8 and 9).

$$F_{1tb} = F_1 \times (1 - f_{1sb}) \quad (8)$$

$$F_{1sb} = F_1 \times f_{1sb} \quad (9)$$

where $F_1$ is the bound fraction obtained from fitting the displacement distributions.

Determination of search dynamics: We estimated the nuclear search kinetics developed by Loffreda et al.[36] We assumed that PcG molecules generally exist in either a bound or a freely diffusing state. We considered a two binding state model (transient vs. stable binding). We assumed that binding to chromatin and unbinding from chromatin are both first-order processes with rate constants $k_{on}^{tb*}$ and $k_{off}^{tb} = \frac{1}{\tau_{tb}}$ for the short-lived PcG molecules, and $k_{on}^{sb*}$ and $k_{off}^{sb} = \frac{1}{\tau_{sb}}$ for the long-lived molecules. $k_{on}^{tb*}$ and $k_{on}^{sb*}$ are pseudo first-order processes, which depend on the concentration of free binding sites.

$$F \underset{k_{off}^{tb}}{\overset{k_{on}^{tb*}}{\leftrightarrows}} N$$

$$F \underset{k_{off}^{sb}}{\overset{k_{on}^{sb*}}{\leftrightarrows}} S$$

where $F$, $N$, and $S$ represents the concentration of free, short-lived, and long-lived molecules, respectively. By considering no dissociation, the probability $f_{1sb}$ was calculated as described by Loffreda et al.[36]

$$\frac{dF^*}{dt} = -k_{on}^{sb*} F^* - k_{on}^{tb*} F^*$$
$$\frac{dN^*}{dt} = k_{on}^{tb*} F^*$$
$$\frac{dS^*}{dt} = k_{on}^{sb*} F^*$$

Thus, we calculated $f_{1sb}$ as follows (Eq. 10).

$$f_{1sb} = \frac{\frac{dS^*}{dt}}{\frac{dS^*}{dt} + \frac{dN^*}{dt}} = \frac{k_{on}^{sb*}}{k_{on}^{sb*} + k_{on}^{tb*}} \quad (10)$$

The target search time for finding a specific site was calculated as follows (Eq. 11).

$$\tau_{search} = N_{trial} \times \tau_{3D} + (N_{trial} - 1)\tau_{tb} \qquad (11)$$

where $N_{trial}$ is the average number of trials which one molecule needs to encounter a specific site, $N_{trial} = \frac{1}{f_{1sb}}$. Please note that the number of trials should be considered as minimum since very brief contacts with chromatin by PcG proteins may not be able to be detected. $\tau_{3D}$ is the average free time between two binding events $\left(\tau_{3D} = \frac{1}{k_{on}^{sb*} + k_{on}^{tb}}\right)$.

By considering the system at equilibrium, the long-lived fraction ($F_{1sb}$) was related to the search time and the residence time.

$$-F \times k_{on}^{tb*} - F \times k_{on}^{sb*} + N \times k_{off}^{tb} + S \times k_{off}^{sb} = 0$$
$$F \times k_{on}^{tb} - N \times k_{off}^{tb} = 0$$
$$F \times k_{on}^{sb*} - S \times k_{off}^{sb} = 0$$

The relationship among the long-lived fraction ($F_{1sb}$), the residence time and the search time was derived as follows (Eq. 12).

$$
\begin{aligned}
F_{1sb} &= \frac{S}{F+N+S} = \frac{k_{on}^{sb*} \times k_{off}^{tb}}{k_{off}^{tb} \times k_{off}^{tb} + k_{on}^{tb} \times k_{off}^{tb} + k_{on}^{sb*} \times k_{off}^{tb}} \\
&= \frac{\left(k_{on}^{sb*} \times k_{off}^{tb}\right) \times \frac{1}{k_{on}^{sb*}+k_{on}^{tb}} \times \frac{1}{k_{off}^{sb} \times k_{off}^{tb}}}{\left(k_{off}^{tb} \times k_{off}^{sb} + k_{on}^{tb} \times k_{off}^{sb} + k_{on}^{sb*} \times k_{off}^{tb}\right) \times \frac{1}{k_{on}^{sb*}+k_{on}^{tb}} \times \frac{1}{k_{off}^{sb} \times k_{off}^{tb}}} \\
&= \frac{\frac{f_{1sb}}{k_{off}^{sb}}}{\frac{1}{k_{on}^{sb*}+k_{on}^{tb}} + \frac{1-f_{1sb}}{k_{off}^{tb}} + \frac{f_{1sb}}{k_{off}^{sb}}} = \frac{\tau_{sb}}{N_{trial} \times \tau_{3D} + (N_{trial}-1) \times \tau_{tb} + \tau_{sb}} \\
&= \frac{\tau_{sb}}{\tau_{search} + \tau_{sb}}
\end{aligned}
\qquad (12)
$$

Thus, the long-lived fraction is determined by the residence time and the search time. If two proteins have the same long-lived fraction, their residence time and search time may be different. In contrast, if the two proteins have different long-lived fractions, their residence time and search time may be the same.

**In vitro single-molecule binding assay**. Preparation of mononucleosome: The K27M mutation in histone H3.3 (human) was introduced by PCR mutagenesis and confirmed by sequencing. Expression and purification of recombinant human histones, and subsequent assembly and purification of histone octamers, followed established procedures[60]. A 216 bp DNA fragment containing the 601 nucleosome positioning sequence was amplified by PCR using pGEM-3z/601 plasmid as the template. The primers used were purchased from IDT (Coralville, IA): /5Alexa488N/ACAGGATGTATATATCTGACACGTGCCTGG and /5Biosg/ TGACCAAGGAAAGCATGATTCTTCACAC. Purified PCR product was used to assemble mononucleosomes by salt dilution[61] and the quality of the nucleosomes were confirmed by native PAGE.

Preparation of JF$_{646}$-labeled Ezh2-PRC2 extract: *FLAG-HaloTag-Ezh2/Ezh2$^{-/-}$* mES cells were grown on two 15 cm gelatin-coated dishes in the presence of 1.0 µg per ml doxycycline. When the confluency reached 95%, cells were collected, resuspended in 2.0 ml of mES cell medium containing 50 nM of HaloTag ligand Janelia Fluor 646 (JF$_{646}$), and incubated at 37 °C in 5% $CO_2$ for 30 min. After centrifugation, cells were washed with 2× pre-chilled PBS buffer. To prepare nuclei, cells were incubated in hypotonic buffer (10 mM HEPES pH 7.9, 1.5 mM MgCl$_2$, 10 mM KCl, and 0.1 mM PMSF) for 10 min on ice, and dounced by glass homogenizer for 12 times. Nuclei were collected at $1000 \times g$ for 3 min at 4 °C and incubated with 500 µl of pre-chilled nuclear lysis buffer containing (20 mM Tris-HCl pH 7.4, 0.05% NP-40, 350 mM NaCl, 0.25 mM EDTA, 20% glycerol, 0.1 mM Na$_3$VO$_4$, and protease inhibitor cocktail (P8340; Sigma-Aldrich) for 30 min. Lysate were collected by centrifugation at $20{,}000 \times g$ for 20 min at 4 °C. To enrich JF$_{646}$-labeled Ezh2-PRC2, we performed one-step enrichment under mild conditions. 75 µl of pre-washed anti-Flag-M2 affinity gel (A2220; Sigma-Aldrich) were incubated with nuclear lysate for 2 h at 4 °C. Protein–bead complexes were washed three times with washing buffer (25 mM Tris-HCl, pH 7.4, 50 mM KCl, 5 mM MgCl$_2$, 1 mM EDTA, 0.5 % NP-40, and 0.5 mg per ml BSA). JF$_{646}$-labeled Ezh2-PRC2 was eluted using elution buffer (washing buffer supplemented with 0.5 mg per ml of 3× FLAG peptide).

Construction of flow cell: Flow cells were constructed according to the reported procedure[55]. A coverslip (48366-249; VWR) was activated by aminosilane (*N*-2-aminoethyl-3-aminopropyltrimethoxysilane (A21541; Pfaltz & Bauer) and then functionalized with biotin PEG-SVA (256-586-9004; Laysan Bio, Arab, AL) and mPEG-SVA (mPEG -SVA-5000; Laysan Bio). The coverslip was assembled to a flow cell with a quartz slide (12-550-15; Thermo Fischer Scientific) having two 0.75 mm holes cross from each other using epoxy glue (14250; Devcon). The assembled flow cell was stored at −20 °C under nitrogen gas.

Imaging by single-molecule microscopy: To each flow cell, 100 µl of solution containing 0.1 mg per ml BSA and 0.2 mg per ml NeutrAvidin (31000; Thermo Fisher Scientific) was loaded and incubated on ice for 15 min. The unbounded NeutrAvidin was removed by flowing 100 µl of T50 buffer (10 mM Tris-HCl pH 8.0, 50 mM NaCl). Mononucleosome labeled with Biotin and Alexa Fluor 488 was loaded into the flow cell and incubated on ice for 10 min. An aliquot of 10 µl of JF$_{646}$-labeled Ezh2-PRC2 extract was added to the flow cell. Images were acquired by TIRF microscopy Zeiss Axio Observer D1 Manual Microscope (Zeiss, Germany) equipped with an Alpha Plan-Apochromatic ×100/1.46 NA Oil Objective (Zeiss, Germany) and an Evolve 512 × 512 EMCCD camera (Photometrics, Tucson, AZ). The pixel size of the EMCCD was 16 µm. Additional ×2.5 magnification was used to generate ×250 overall magnification. Alexa Fluor 488 and JF$_{646}$ were excited at 488 nm and 640 nm, respectively, using a solid state laser stack (Intelligent Imaging Innovations). A TIRF Laser Microscope Cube for 488/640 Excitation (C182369; Chroma) was used to filter the excitation wavelength. A laser power intensity of ~10 mW per cm$^2$ was used for exciting Alexa Fluor 488 and a power intensity of ~20 mW per cm$^2$ for exciting JF$_{646}$. The microscope and the EMCCD camera were controlled by Slidebook 6.0 software. The position of nucleosome was recorded first with a 200-ms exposure time, and then JF$_{646}$ was track for 1000 frames with a 30-ms integration time and 470-ms dark time. Finally, the position of nucleosome was recorded again.

Data analysis: Data analysis was performed by using ImageJ (http://imagej.nih.gov/ij/). We colocalized the positons of nucleosomes before and after imaging of JF$_{646}$-labeled Ezh2-PRC2 and discarded images where the positions do not colocalize. About 80% of image stacks were discarded. We considered JF$_{646}$-labeled Ezh2-PRC2 molecules as binding to nucleosomes only if they colocalize with nucleosome. The time traces of fluorescent intensity of JF$_{646}$-labeled Ezh2-PRC2 molecules were generated by ImageJ. The dwell time of individual JF$_{646}$-labeled Ezh2-PRC2 molecules were directly measured as the lifetime of the fluorescence spots. The survival distribution of dwell time was necessary to fit with one-component decay function based the F-test implemented in OriginLab.

$$y(\tau) = Ae^{(-\tau/\tau_0)} \qquad (13)$$

where $\tau_0$ is residence time and $A$ is amplitude. Please note that the estimated residence time is at the lower limitation since the photobleaching has not been corrected.

**Data availability**. All relevant data are available from the authors on request. https://figshare.com/articles/NCommunications-4-21-2018_xlsx/6169016

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

## Acknowledgements

We thank Dr. Haruhiko Koseki for providing *Eed*⁻/⁻ mES cell lines, Drs. Stuart Orkin and Xiaohua Shen for providing *Ezh2*⁻/⁻ mES cell line, and Dr. Luke D. Lavis for providing Janelia Fluor® Dyes. This work was supported, in whole or in part, by the National Cancer Institute under Award Number R03CA191443 (to X.R.) and NIH R01CA 204297 (to Z.Z.), the National Science Foundation under Award Number CHE-1500285 (to H.W.), and the Office of Research Services (ORS) at University of Colorado Denver.

## Author contributions

R.T. established transgenic mESC lines, performed live-cell SMT, in vitro-binding assay, immunofluorescence, and data analysis. T.N.H. constructed plasmids, established transgenic mESC lines, performed the killing curve and immunofluorescence, and data analysis. H.N.D. constructed plasmids, established mESC lines, and performed data analysis. B.S. and Y.T. provided reconstituted nucleosomes. F.D. and Z.Z. provided cell lines and plasmids. X.S. and Y.D. wrote Matlab script. C.P. provided analytic tools. H.W.

performed data analysis and supervised students. X.R. conceived and designed the study, supervised and performed the experiments, analyzed data, prepared the figures, and wrote the paper. All authors provide constructive comments for the manuscript.

## Additional information

**Competing interests:** The authors declare no competing interests.

