## [Peer Review File · Nature Communications]

Reviewers' comments:

Reviewer #1 (Remarks to the Author):

Review of “Live-cell Single-Molecule Dynamics of PcG Proteins Imposed by the DIPG H3.3K27M Mutation”

Overall assessment

This paper from Tatavosian and Huynh from Xiaojun Ren’s laboratory focuses on the dynamics of Polycomb complex 1 and 2 building on their previous work (Zhen et al, eLife 2016), but here focusing specifically on the glioma disease relevant H3.3K27M mutation and how it affects Polycomb (PcG) dynamics.

Overall, an impressive number of techniques and systems are brought together to study this important question: they study both mES and HEK cells including mES cells KO for Ezh2 and Eed using sophisticated single-molecule imaging both inside cells and in vitro.

They find the H3.3K27M mutation has distinct effects on PRC1 and PRC2 (bound fraction, residence time) and also find that over-expressing PRC1 reduces proliferation of glioma cells. The manuscript is also generally well-written and clear (with a few exceptions, see below) and the data is generally honestly and well-presented and clearly described. However, although the study is generally very well done, interesting and clearly presented, I was very surprised to see the analysis of diffusion dynamics (logD). Not only has this approach been known to be highly flawed for many years, the methods section pertaining to this also contains a couple of clearly false statements.

Nevertheless, once the authors fix these issues (which will require major re-analysis, but should not take too long since new experiments are not necessary), this will be a very nice story suitable for publication that is likely to gather significant interest and be widely read.

MAJOR REVISIONS

Tracking diffusing molecules

To calculate the different populations (bound, confined, free diffusion) the authors assign diffusion constants to each track and then fit the log₁₀ distribution of fitted diffusion constants with a 2 or 3 Gaussians. No theoretical validation or justification is given beyond saying that REF10, 49-52 also did it and the fact that other people did their analysis incorrectly does not justify continuing. The number of independent papers that have shown that this approach is incorrect is long and I cannot give a full list here. Here are 2:

Michalet and Berglund (Phys. Rev. E. 85, 061916) showed in 2012 that extremely long

trajectories are necessary to accurately estimate D . In live-cell SMT, authors are 1-2 orders of magnitude from the Michalet/Berglund limit and the inferred D is unreliable.

Weimann, Ganzinger, ..., Kleinerman showed in 2013 (PLOS ONE e64287) that even for membrane protein (which diffuse slowly, in 2D and give very long trajectories), the MSD/LogD approach is inaccurate. And this was with very long trajectories.

It is especially strange that the authors choose the MSD approach since the Jump Distance approach is seemingly not more complicated to implement. For a full displacement model please see Mazza 2012.

This brings us to a 2nd point. The authors claim that 5-20% of molecules move out of the focal plane (line 475-476). This is obviously wrong – and the statement makes no sense without saying over what time and what D . For molecules with $D > 2$, if you wait > 100 ms, most molecules have moved out of the focal plane unless your focal depth is many microns. What you need to do, is to correct as a function of time and diffusion constant. See Kues+Kubitscheck 2002 for how to do this ([http://onlinelibrary.wiley.com/doi/10.1002/1438-5171\(200208\)3:4<218::AID-SIMO218<3.0.CO;2-C/full](http://onlinelibrary.wiley.com/doi/10.1002/1438-5171(200208)3:4<218::AID-SIMO218<3.0.CO;2-C/full)). It does involve integrating an infinite sum, but it is a necessary correction. If the authors have trouble implementing this, there was a recent methods paper making code available for doing this (<https://www.biorxiv.org/content/early/2017/08/22/171983>; “Spot-On”). Incidentally, this paper also shows that the MSD/LogD approach taken by the authors in this paper is generally unreliable.

The authors claim that the jump distance analysis gave the same results in Fig S3, but they do not show anywhere I could find what parameters they get out of the displacement analysis (e.g. bound fraction); also they do not show how they did fitting to the displacement CDF (e.g. which equation)? Even if this is true in a few cases that the displacement and MSD approach gives similar results, the MSD approach is still wrong and perpetuating this will hurt the field.

Therefore, the authors should re-analyze their data used for LogD analysis (e.g. Fig. 1C-E, 3B and so on) using displacement analysis - they can either re-do their displacement analysis with the Kues&Kubitscheck correction implemented or they can just use the publically available Spot-On user interface.

But before I can recommend publication, the LogD analysis should be removed from the main figures and replaced with proper displacement analysis with out-of-focus correction (since code to re-do the analysis is freely available (Spot-On), this should not take too long).

Finally, the whole SMT bias discussion in lines 470-482 is mostly wrong and the authors should re-write it and honestly correct their mistakes (5-20% out of focus, this is wrong; 0.6% of molecules move > 2 pixels, this is wrong – just looking at the authors own movies clearly show that many molecules move more than 2 pixels). The authors must honestly acknowledge in the main text that they cannot exclude some bias from “diffusion blur”. Also, the fact that they get a HaloTag-NLS $D = 3.6 \text{ um}^2/\text{s}$ (Table S1), which is much smaller than other reports, suggest that “diffusion blur” is strongly limiting their SMT. Instead of fudging things, the authors should just honestly acknowledge this limitation in the main text.

MINOR REVISIONS

Diffusion blurs

The authors discuss bias from “diffusion blur”, but their discussion is mostly wrong. Instead of claiming things that are wrong, the authors should just honestly state that they couldn’t exclude some bias from “diffusion blur”. The authors claim that only 0,6% of molecules move more than 2 pixels and therefore, that “diffusion blur” is minimal. How did they calculate this? See Mazza 2012 page 3 bottom left for an equation for 2D diffusion. If you slightly re-write it you get: $P(\text{move more than } 2 \times 64 \text{ nm}) = \exp(-r_{\text{MAX}}^2 / 4 * D * t) = \exp(- (0.128 \text{ um})^2 / (4 * 2.5 \text{ um}^2/\text{s} * 0.030 \text{ s})) = 94.69\%$. In other words, 94.69% of all freely diffusing molecules move more than 2 pixels (using the authors numbers: 30 ms, $D=2.5$, $r_{\text{MAX}} = 2*64 \text{ nm}$). How the authors manage to get this calculation so wrong is unclear, but if anything the calculation suggests that “diffusion blur” massively biases their data. The authors should re-write this section entirely.

Search dynamics

I think the authors maybe slightly over-interpret their search mechanism (number of trials) etc. Most likely, nuclear proteins make many contacts that are too brief to be seen by SMT, so it is a bit arbitrary to say that only non-specific events lasting long enough to be observed are called non-specific. But anyway, this is a small point and the authors can keep this description the way they judge is best.

Mechanism

The authors elude to how H3K27M might effect PcG binding in line 301. I understand that there has been previous structural work, but a bit more info and/or speculation on the mechanism could be interesting to hear in the discussion section.

Misc

mES cells: many different ones were used, please give strain background for each.

SMT imaging: authors say 15mW, but units should be mW/cm² should they not?

U-TRACK: authors say they use U-TRACK, but they must give the parameters they used in this algorithm (all of the parameters to ensure reproducibility).

Determination of search dynamics (line 499-515)

The authors clearly describe their approach, but I think saying that “We estimated the nuclear search kinetics introduced by Liu et al. 31 and further developed by Loffreda et al 33.”, is very unfair. Not only did Liu/Chen 2014 get the calculations wrong, they also published numbers that were wrong and they have not corrected their mistakes so that several new papers still use their incorrect approach. Loffreda thankfully fixed this. To avoid perpetuating the incorrect approach, the authors should either only cite Loffreda or they should explicitly say that Loffreda corrected

Liu/Chen's mistakes.

Writing

The paper is generally well-written but contains several sentences that are unclear or grammatically wrong. Here are some:

-Abstract: "has no effect on the chromatin-bound level": it's a bit unclear. Maybe say fraction bound to chromatin.

LINE NUMBERS:

-79-80: the statement about clinical implications is too strong. Please tone down.

-123-124: grammar. "May make ... cannot"

-138: "dynamic"

-246-246: bad syntax

-261: cannot "become" less proliferation

Reviewer #2 (Remarks to the Author):

The manuscript by Tatavosian et al. describe the application of state-of-the-art live-cell single-molecule tracking approach to characterize the search mechanism of Polycomb subunits for target sites in wt-mES cells and in cells expressing a mutated H3 (H3.3K27M) variant, that affects H3.3k27 methylation in gliomas. The authors find that this mutation differently affect the search of different pCG proteins: The binding of the catalytic subunit of PRC2, Ezh2, is found to be stabilized in conditions when H3.3k27M is expressed), suggesting sequestration of Ezh2.

Differently the frequency of binding of CBX7, the chromodomain protein of PRC1 and a "reader" of the of H3 modifications is reduced in H3.3k27M expressing cells, in agreement with the general reduction in H3.3k27me3 in these conditions.

The part of the manuscript relating to Ezh2, aims to resolve conflicting biochemical data on the role of H3.3Me mutant in PRC2 targeting, and I feel these novel results would be very interesting for the researchers working on the epigenetics of pediatric gliomas.

Differently, the results obtained on CBX7 are of little surprise as the authors have shown previously (Zhen et al. Elife, 2016), that PRC1 targeting depends on the recognition of H3.3K27Me3, and this modification is greatly reduced in cells carrying a copy of mutated H3.3. Nevertheless, the relevance of this mutation in pediatric cancer renders these results of potential

great interest for the field.

More globally, the paper is a first example about the role of how epigenetic alterations in cancer can divert the search mechanism of chromatin modifiers, extending the scope of action of intranuclear single-molecule approaches that so far mostly focused on transcription factors.

Therefore I believe that the paper is a good candidate for publication in Nature Communications. However, I have some concerns on how bound fractions and search times are calculated by the authors and in my opinion these biases might undermine the interpretation of the results (see below). Also I have some concerns/comments on other technical and biological details that I would like the authors to address.

MAJOR CONCERN: ANALYSIS OF THE DISTRIBUTION OF DIFFUSION COEFFICIENTS and SEARCH TIME CALCULATION.

If I understood right, the D_m calculation is track based (it is basically extracted from the first point of the time-averaged mean squared displacement curve), and each measured diffusion coefficient is not weighted for the duration of the track. I believe that it would be fine to use this method if the authors could follow molecules in the whole 3D volume of the cell. However, due to the limited thickness of the observation slice, I believe that that the method used can result in strong underestimation of the bound fractions, with a bias that depends on the duration of the binding events.

To explain this I have to use an example. First of all, let's start from the definition of bound fraction that the authors also use at page 20, line 511: $F_1 = t_{\text{average}} / (t_{\text{average}} + t_{3d})$. This is analogous (for ergodicity) to define F_1 as the fraction of molecules being bound at every SINGLE TIME.

Let's now consider the case in which a molecule binds for a very long time, and stays unbound for the same amount of time (so it's bound fraction should be 50%), and let's assume that you are in a condition in which in your detection slice only two molecules are visible at each time. The bound molecule will always be the same one through-out the whole movie and therefore will contribute with only one count to the distribution of D_m . Instead the free molecules will be in constant exchange, for every molecule that leaves the detection slice (typically in a couple of frames), a second one will jump in (the system is at equilibrium) and therefore the free molecules will contribute with MANY counts on the histogram of D_m s, EVEN IF at every single time you just have one free and one bound molecule in the image. The problem is very evident for the H2A movie the authors provide as SI. In each frame most of the molecules (I would say more than 90%) appear as bound with only one or two molecules diffusing around. Therefore one would expect to measure a bound fraction higher than 90%. However, due to the fact that bound molecules are counted only once, independently on the number of frames for which the molecules appear as bound, the bound fraction estimated by the author's method is much lower

(let's than 70%).

Importantly, I believe that the alternative way of analyzing the data proposed by the authors in the supplementary information (looking at just the first displacement of each track) would be affected by the same bias, and therefore an alternative strategy should be implemented to correctly analyze bound fractions.

One trivial way to correct this issue would be to state that the measured bound fractions represent the fraction of TRACKS (rather than molecules) in the bound state. Unfortunately this approach would fail for the measurements of search times, as this calculation requires to measure the TRUE bound fraction (that is the fractional time a molecule stays in the bound state).

Approaches that could be used to correctly calculate the bound fraction are the mathematical model derived in (Mazza et al, Nucl Ac Res, 2012), or the simplified methods used in either (Loffreda et al., Nat Comm, 2017) or (Hansen et al. eLife, 2017). Recently, Dr. Darzacq group has developed a web tool that allows to fit the tracking data with the last listed method (Hansen et al Biorxiv, 2017). In the paper describing the interface, the authors highlight the risks of misestimating bound fractions using a track-based approach in HiLo SMT of nuclear factors.

Given the little detail provided in how bound fractions are estimated I might have missed something in how the authors analyzed their tracking data. In this case I apologize for my comment, and I suggest the authors to better explain how they account for the different duration of the free and bound track segments.

OTHER TECHNICAL COMMENTS:

1. The authors refer to long-lived and short lived binding as the result of bi-exponential fitting of the 1-CDF distribution of residence times arising from interleaved single molecule imaging at low frame rates. Next, the parameters arising from the fit are used to compute search times according to (Loffreda et al Nat. Comm, 2017). However, these calculations underlie the assumption that the long-lived interactions are the one that are biologically relevant (the “specific binding” to cognate sites). In the transcription factors SMT literature these assumptions are typically tested by imaging mutants with impaired specific binding (Gebhardt et al., Nat Methods, 2013, Chen et al., Cell 2014 and so on). What is the evidence for PRC1 and PRC2 members that long lived binding is representative of binding to cognate sites? I recall that some of these tests were performed on Cbx7 in their previous paper, but it would be useful to show that an incompetent Ezh2 has a reduced fraction of molecules engaged in long-lived binding. The EED KO experiment is somewhat arguing towards that direction, however to me it is puzzling that upon EED depletion, the search times of Ezh2 decrease and the authors should clarify the meaning of this result.

2. The authors use quite long exposure times (30ms) to track molecules, and in this situation motion blur might affect the estimates of BOTH the fractions of bound/free molecules and the estimation of the diffusion coefficients itself (as fast diffusing molecules will be systematically more affected by motion blur compared to slow ones). The effect has been described in several papers (Izeddin et al., eLife, 2014; Hansen et al., eLife, 2016). In the methods sections, the authors recognize this potential issue, however, in my opinion they underestimate the effect that such bias could have on their measurements. As the estimation of the diffusion coefficients might be affected by motion blur as well, different proteins might have different REAL diffusion coefficients, and therefore the bias favoring bound molecules might have different magnitudes in the different conditions tested. While the effect might not be dramatic, it still needs to be tested. Typically, motion blur is minimized by using shorter laser exposure times (< 5 ms), potentially via stroboscopic illumination. I suggest the authors to try this approach in a few key conditions (e.g. Cbx7 in WT and H3.3k27M expressing cells), to cross-validate their results. Also, for the problems mentioned above, currently the discussion on the differences in diffusion coefficients (for example lines 109-113 in the results) could be misleading.

GENERAL COMMENTS ON THE BIOLOGICAL QUESTIONS

1. Sequestration of PRC2. As the authors report in the introduction, there is an unresolved issue in the literature as some reports show enrichment of PRC2 on H3K27M containing nucleosomes, while more recent studies show no evidence of PRC sequestration at these sites by ChIP-seq. The in-vitro single molecule data of the authors provide compelling evidence that some sequestration occurs. Similarly, the in-vivo SMT data shows that on average Ezh2 binding lasts 1.5x longer when H3K27M is expressed, albeit it is currently not clear if this longer binding occurs at loci enriched for this mutated histone.

I feel that in the current form of the manuscript it is still not clear whether the authors believe that PRC2 is sequestered in cis- at H3K27M sites or if the epigenetic reprogramming caused by H3K27M is responsible for diverting PRC2 search. The authors could try to restate the relevant parts of the discussion to better clarify how their data fit in the current models for PRC2 inhibition in pediatric glioma.

Ideally some additional data could help in clarifying the underlying mechanism.

- Did the authors tried to use a fluorescent version of H3K27M (for example by tagging it with GFP) to show whether this mutant localizes in discrete foci in the nucleus? If yes, the authors could try to measure whether the residence time of Ezh2 is indeed longer at these foci compared to the rest of the nucleoplasm using a single molecule approach such as the one previously described in (Morisaki et al., Nat. Comm., 2014).

- For the "global reprogramming" hypothesis: is there any ChIPSeq data showing that the position of PRC2 binding sites are widely different in cells expressing H3.3k27M compared to WT? If not, would it be possible to generate these kind of data?

I am aware that these experiments might extend beyond the scope of the paper. The authors should therefore, at minimum, restate the discussion to specify that future experiments are needed to clarify the mechanism underlying the control of PRC2 search by H3.3k27Me.

2. CBX7 over-expression and DIPG cells proliferation. The results of Figure 6 are interesting although not completely novel, as (Yu et al., Oncotarget, 2017) have recently shown that Cbx7 is a prognostic marker for gliomas and its overexpression result in reduced proliferation. This paper also show a hint about the mechanism by which Cbx7 overexpression could promote cell cycle arrest, by binding and repressing the Cyclin E1 promoter.

As presented, the r results do not clarify how the decrease in proliferation relates to the search mechanism of CBX7. At minimum, the authors should show the effect of tuning the expression levels of CBX7 on its search mechanism. Ideally the CBX7 search time should remain unchanged, or (even better) decrease. Either of these two situations would mean that any given time more target sites are bound by CBX7 when over expressed. This observation could be then reconciled with the data from Yu et al.

TO SUM UP:

The authors show interesting modulation of PcG protein search mechanisms depending on the expression of mutated histones. I have some concerns on the way the SMT data has been analyzed, and if the main results hold true after further validation, the paper is in my opinion of high interest. However, the authors should better define how Cbx7 overexpression affects its search mechanism and to clarify their model for PRC2 inactivation in cells expressing H3K27M.

Minor Points:

- p. 10 Last sentence of results. “Proliferation” should be changed in “proliferating”
- p. 16 Line 401. The range of dye concentration used should be specified.
- P. 16. Line 418. The laser power is expressed in mW. However it would be more useful to provide the irradiance, given by the laser power divided by the area of the field of view illuminated by the laser – so that others could try to replicate the settings.

Reviewer #3 (Remarks to the Author):

The Histone H3 K27M missense mutation is a frequently found genetic alternation in pediatric diffuse intrinsic pontine glioma (DIPG) and other midline high-grade gliomas. The K27M mutant histone leads to a loss of H3K27me_{2/3} apparently via inhibition of the polycomb repressive complex 2 (PRC2). Despite global loss of K27me_{2/3}, focal gains of K27me₃ were

noted in K27M-containing cell lines. How these K27me3 foci contributed to K27M-mediated tumorigenesis remains unknown.

Here, the authors used a live-cell single molecule tracking system to explore diffusion of PRC1 (CBX7) and PRC2 (EZH2 and EED) subunits in murine embryonic stem cells. Based on SMT measurements, the authors conclude that different populations (chromatin-bound, confined diffusion, free diffusion) exist for PRC1 and PRC2 complexes. Residence time of chromatin-bound complexes was also assessed. The authors find that cells expressing the K27M histone alter the diffusion kinetics of CBX7, but not PRC2 subunits in HEK293T cells. While the diffusion kinetics of PRC2 were not affected in K27M-expressing HEK293T cells, oddly, the authors did find that K27M affected the diffusion of both EZH2 and CBX7 in DIPG cells. In support of findings from previous studies, the authors find that K27M stabilizes EZH2 on nucleosomes *in vitro*. The authors find that overexpression of CBX7 suppresses DIPG cell growth *in vitro*.

The observation that the K27M mutation affects PRC2 and PRC1 localization has been noted in previous studies using high resolution ChIP sequencing techniques. The observation that overexpression of CBX7 suppresses growth of DIPG cells is potentially interesting if the mechanism could be teased out. Overall, the manuscript has some interesting observations, but lacks mechanistic insight into how the presence of K27M histones leads to an increase in PRC2 activity at specific genic loci. Also, it does little to explain how a mutant histone leads to a unique chromatin state that promotes tumorigenesis. The manuscript relies too much on only using the SMT technique and fails to use orthogonal approaches to support conclusions based on SMT data. In summary, in the absence of any new mechanistic data about PRC2, or how K27M leads to gain/loss of K27me3, I believe that this manuscript does not provide sufficient novel information to justify publication in Nature Communications.

Other comments:

Need to define the term 'SMT' in the paper. The term is presented without explanation.

Line 110 "Ezh2 and Eed dynamically exchange"-- too strong without additional data supporting this finding. Include "may or might dynamically exchange".

Responses to Reviewers' Comments:

We are grateful to the Reviewers for their insightful critiques on our manuscript. We have re-analyzed all the data and performed additional experiments to address their critiques. In summary, based on our data and literature, we now propose that the residence time of PcG proteins directly correlates with their activities and the target search time of PcG proteins is critical for regulating their genomic occupancy.

Please note that the comments of the Reviewers are quoted verbatim using *italic and blue type*. Our rebuttal is in black.

Reviewers' comments:

Reviewer #1 (Remarks to the Author):

Review of "Live-cell Single-Molecule Dynamics of PcG Proteins Imposed by the DIPG H3.3K27M Mutation"

Overall assessment

1.1) This paper from Tatavosian and Huynh from Xiaojun Ren's laboratory focuses on the dynamics of Polycomb complex 1 and 2 building on their previous work (Zhen et al, eLife 2016), but here focusing specifically on the glioma disease relevant H3.3K27M mutation and how it affects Polycomb (PcG) dynamics. Overall, an impressive number of techniques and systems are brought together to study this important question: they study both mES and HEK cells including mES cells KO for Ezh2 and Eed using sophisticated single-molecule imaging both inside cells and in vitro. They find the H3.3K27M mutation has distinct effects on PRC1 and PRC2 (bound fraction, residence time) and also find that over-expressing PRC1 reduces proliferation of glioma cells. The manuscript is also generally well-written and clear (with a few exceptions, see below) and the data is generally honestly and well-presented and clearly described. However, although the study is generally very well done, interesting and clearly presented, I was very surprised to see the analysis of diffusion dynamics (logD). Not only has this approach been known to be highly flawed for many years, the methods section pertaining to this also contains a couple of clearly false statements. Nevertheless, once the authors fix these issues (which will require major re-analysis, but should not take too long since new experiments are not necessary), this will be a very nice story suitable for publication that is likely to gather significant interest and be widely read.

We thank the Reviewer for their constructive critiques and their insights offered. We have implemented their suggestions by performing additional experiments and re-writing the manuscript.

MAJOR REVISIONS

Tracking diffusing molecules

1.2) To calculate the different populations (bound, confined, free diffusion) the authors assign diffusion constants to each track and then fit the log10 distribution of fitted diffusion constants with a 2 or 3 Gaussians. No theoretical validation or justification is given beyond saying that REF10, 49-52 also did it and the fact that other people did their analysis incorrectly does not justify continuing. The number of independent papers that have shown that this approach is incorrect is long and I cannot give a full list here. Here are 2: Michalet and Berglund (Phys. Rev. E. 85, 061916) showed in 2012 that extremely long trajectories are necessary to accurately estimate D. In live-cell SMT, authors are 1-2 orders of magnitude from the Michalet/Berglund limit and the inferred D is unreliable Weimann, Ganzinger, ..., Kleinerman showed in 2013 (PLOS ONE e64287) that even for membrane protein (which diffuse slowly, in 2D and give very long trajectories), the MSD/LogD approach is inaccurate. And this was with very long trajectories. It is especially strange that the authors choose the MSD approach since the Jump Distance

approach is seemingly not more complicated to implement. For a full displacement model please see Mazza 2012. This brings us to a 2nd point. The authors claim that 5-20% of molecules move out of the focal plane (line 475-476). This is obviously wrong – and the statement makes no sense without saying over what time and what D. For molecules with $D > 2$, if you wait > 100 ms, most molecules have moved out of the focal plane unless your focal depth is many microns. What you need to do, is to correct as a function of time and diffusion constant. See Kues and Kubitscheck 2002 for how to do this ([http://onlinelibrary.wiley.com/doi/10.1002/1438-5171\(200208\)3:4%3C218::AID-SIMO218%3E3.0.CO;2-C/abstract](http://onlinelibrary.wiley.com/doi/10.1002/1438-5171(200208)3:4%3C218::AID-SIMO218%3E3.0.CO;2-C/abstract)). It does involve integrating an infinite sum, but it is a necessary correction. If the authors have trouble implementing this, there was a recent methods paper making code available for doing this (<https://www.biorxiv.org/content/early/2017/08/22/171983>; “Spot-On”). Incidentally, this paper also shows that the MSD/LogD approach taken by the authors in this paper is generally unreliable. The authors claim that the jump distance analysis gave the same results in Fig S3, but they do not show anywhere I could find what parameters they get out of the displacement analysis (e.g. bound fraction); also they do not show how they did fitting to the displacement CDF (e.g. which equation)? Even if this is true in a few cases that the displacement and MSD approach gives similar results, the MSD approach is still wrong and perpetuating this will hurt the field. Therefore, the authors should re-analyze their data used for LogD analysis (e.g. Fig. 1C-E, 3B and so on) using displacement analysis - they can either re-do their displacement analysis with the Kues&Kubitscheck correction implemented or they can just use the publically available Spot-On user interface. But before I can recommend publication, the LogD analysis should be removed from the main figures and replaced with proper displacement analysis with out-of-focus correction (since code to re-do the analysis is freely available (Spot-On), this should not take too long).

We agree that the displacement analysis is based on a rigorous kinetic framework. We have re-analyzed all the data based on the displacement histogram using Spot-On (Hansen et al. Elife 7(2018)), which is based on a kinetic modelling framework first described by Mazza et al. (Mazza et al. Nucleic Acids Res 40, e119 (2012)). The Spot-On analysis accounts for the defocalization bias. The new analysis does not change the conclusions of our manuscript. We have replaced all the LogD analysis with the Spot-on analysis in the revised manuscript.

In the original manuscript, we made mistakes in calculating % of molecules jumping out of focus. The accumulative probability of molecules remaining within focus should be $(1 - \exp(-\frac{r^2}{4Dt}))$. In the original manuscript, the % of molecules reported should be within the focus. We have corrected this in the revised manuscript. We apologize for the mistake.

We have re-written the entire section of “Extraction of PcG bound fractions and diffusion constants” (p. 18-19 line 482-511).

1.3) *Finally, the whole SMT bias discussion in lines 470-482 is mostly wrong and the authors should re-write it and honestly correct their mistakes (5-20% out of focus, this is wrong; 0.6% of molecules move > 2 pixels, this is wrong – just looking at the authors own movies clearly show that many molecules move more than 2 pixels). The authors must honestly acknowledge in the main text that they cannot exclude some bias from “diffusion blur”. Also, the fact that they get a HaloTag-NLS $D = 3.6 \text{ } \mu\text{m}^2/\text{s}$ (Table S1), which is much smaller than other reports, suggest that “diffusion blur” is strongly limiting their SMT. Instead of fudging things, the authors should just honestly acknowledge this limitation in the main text.*

We agree that both “motion blur” and “defocalization” lead to an overestimation of bound molecules. In the original manuscript, we made mistakes in these calculations (please also see the reply 1.2). The accumulative probability of a molecule that starts at the origin at time 0 and finds within distance r should be $(1 - \exp(-\frac{r^2}{4Dt}))$. We have corrected the mistake in the revised manuscript.

One of the nice features of Spot-On is that it corrects the “defocalization” effect. In the revised manuscript, we made statement saying that “One bias is that 3D freely diffusing molecules will jump out of the Z axial detection window (Kues et al. Single Molecules 3, 218-224 (2002)), which has been corrected by the Spot-On analysis (Hansen et al. Elife 7(2018))” (p. 19 line 499).

To test whether “motion blur” affects our conclusion, we have performed a series of experiments (Ezh2 and Cbx7 in *H3.3-* and *H3.3K27M*-expressing cells; and Ezh2 and Cbx7 in 9427 and SF8628 cells) by varying exposure times 5 ms, 10 ms, 20 ms, and 30 ms *via* stroboscopic illumination (Fig S4-5). These results indicate that motion blur results in a slight overestimation of bound fraction (Table S1); however, the overall conclusions remain unchanged. In the revised manuscript, we made the following statement (p. 19 line 500-508).

“Another bias is that 3D freely diffusing molecules tend to “motion blur” because they can move several pixels during a short exposure time. The extent of the bias by motion blur is sensitive to the experimental conditions and the localization algorithm (Hansen et al. Elife 7(2018)). To test whether motion blur influences our conclusions, we performed a series of experiments in which we fixed the lag time (30 ms), but varied the exposure times (5 ms, 10 ms, 20 ms and 30 ms) (Fig S4-5). Motion blur caused a slight overestimation of bound PcG proteins (Table S1), but did not affect our conclusions. It should be noted that even at less than 5 ms exposure time, motion blur still contributes to the overestimation of bound PcG proteins. However, our optic setup limits us to further reduce the exposure time due to signal-to-noise.”

MINOR REVISIONS

Diffusion blurs

1.4) *The authors discuss bias from “diffusion blur”, but their discussion is mostly wrong. Instead of claiming things that are wrong, the authors should just honestly state that they couldn’t exclude some bias from “diffusion blur”. The authors claim that only 0.6% of molecules move more than 2 pixels and therefore, that “diffusion blur” is minimal. How did they calculate this? See Mazza 2012 page 3 bottom left for an equation for 2D diffusion. If you slightly re-write it you get:*

*$P(\text{move more than } 2 \times 64 \text{ nm}) = \exp(-r_{\text{MAX}}^2 / 4 * D * t) = \exp(- (0.128 \text{ um})^2 / (4 * 2.5 \text{ um}^2/\text{s} * 0.030 \text{ s})) = 94.69\%$. In other words, 94.69% of all freely diffusing molecules move more than 2 pixels (using the authors numbers: 30 ms, $D=2.5$, $r_{\text{MAX}} = 2 * 64 \text{ nm}$). How the authors manage to get this calculation so wrong is unclear, but if anything the calculation suggests that “diffusion blur” massively biases their data. The authors should re-write this section entirely.*

We apologize for the mistake for these calculations. We have corrected the mistake in the revised manuscript. Please also see the reply 1.2 and 1.3. To investigate whether motion blur affects our conclusions, we have performed a series of experiments (Ezh2 and Cbx7 in *H3.3-* and *H3.3K27M*-expressing cells; and Ezh2 and Cbx7 in 9427 and SF8628 cells) by varying exposure times 5 ms, 10 ms, 20 ms, and 30 ms *via* stroboscopic illumination (Fig S4-5). These results indicate that motion blur results in a slight overestimation of bound fraction (Table S1), but does not affect our conclusions. We have updated the text accordingly (p. 19 line 500-508).

Search dynamics

1.5) *I think the authors maybe slightly over-interpret their search mechanism (number of trials) etc. Most likely, nuclear proteins make many contacts that are too brief to be seen by SMT, so it is a bit arbitrary to say that only non-specific events lasting long enough to be observed are called non-specific. But anyway, this is a small point and the authors can keep this description the way they judge is best.*

We agree with their comments. In the revision, we added that “Please note that the number of trials should be considered as minimum since very brief contacts with chromatin by PcG proteins may not be able to be detected” (p. 20 line 543).

Mechanism

1.6) *The authors elude to how H3K27M might effect PcG binding in line 301. I understand that there has been previous structural work, but a bit more info and/or speculation on the mechanism could be interesting to hear in the discussion section.*

We thank the Reviewer for their comments. We have re-analyzed all the data and performed additional experiments to provide the mechanisms that underpin how H3.3K27M inhibits the trimethylation activity of PRC2 *in vitro* and reprograms the genomic occupancy of H3K27me3 and PRC2 *in vivo*. We also provided insights into the binding and search kinetics of Cbx7 regulated by increasing its protein levels. We have re-written the entire section of Discussion accordingly (p. 12-14). In summary, our analyses indicate that the target search time plays a critical role in regulating the genomic occupancy of PcG proteins and the residence time directly correlates with the activities of PcG proteins.

Misc

1.7) *mES cells: many different ones were used, please give strain background for each.*

In the revised manuscript, we reported strain background for each mES lines (p. 15 line 386-389).

1.8) *SMT imaging: authors say 15mW, but units should be mW/cm² should they not?*

Thanks for the suggestion. We reported laser power density with mW/cm² (p. 17 line 472 and p. 23 line 606).

1.9) *U-TRACK: authors say they use U-TRACK, but they must give the parameters they used in this algorithm (all of the parameters to ensure reproducibility).*

In the revised manuscript, Table S2 includes the u-track parameters used in this research.

1.10) *Determination of search dynamics (line 499-515)*

The authors clearly describe their approach, but I think saying that “We estimated the nuclear search kinetics introduced by Liu et al. 31 and further developed by Loffreda et al 33.” is very unfair. Not only did Liu/Chen 2014 get the calculations wrong, they also published numbers that were wrong and they have not corrected their mistakes so that several new papers still use their incorrect approach. Loffreda thankfully fixed this. To avoid perpetuating the incorrect approach, the authors should either only cite Loffreda or they should explicitly say that Loffreda corrected Liu/Chen’s mistakes.

We understood this. In the revised manuscript, we made statement saying that “We estimated the nuclear search kinetics developed by Loffreda et al. (Loffreda et al. Nat Commun 8, 313 (2017))” (p. 20 line 529).

Writing

1.11) *The paper is generally well-written but contains several sentences that are unclear or grammatically wrong. Here are some:*

-Abstract: "has no effect on the chromatin-bound level": it's a bit unclear. Maybe say fraction bound to chromatin.

We revised it as "its fraction bound to chromatin".

LINE NUMBERS:

1.12) *79-80: the statement about clinical implications is too strong. Please tone down.*

We have revised the entire paragraph (p. 4 line 67-72).

1.13) *123-124: grammar. "May make ... cannot"*

We revised the original sentence as "Since the HaloTag-PRC2 subunit fusions and their endogenous counterparts co-exist within cells, the fusion proteins may be at disadvantage to compete with their endogenous counterparts for binding sites" (p. 5 line 104).

1.14) *138: "dynamic"*

We have corrected "dynamic" as "dynamics".

1.15) *246-246: bad syntax*

We revised the original sentence as "The cumulative frequency distribution of dwell times was fitted with a one-component exponential decay function" (p. 10 line 261).

1.16) *261: cannot "become" less proliferation*

Thanks for carefully reviewing the manuscript. We revised the original sentence as "These results demonstrate that human DIPG cells grow slowly in response to increasing Cbx7 expression" (p. 11 line 274).

Reviewer #2 (Remarks to the Author):

2.1) The manuscript by Tatavosian et al. describe the application of state-of-the-art live-cell single-molecule tracking approach to characterize the search mechanism of Polycomb subunits for target sites in wt-mES cells and in cells expressing a mutated H3 (H3.3K27M) variant, that affects H3.3K27 methylation in gliomas. The authors find that this mutation differently affects the search of different PcG proteins: The binding of the catalytic subunit of PRC2, Ezh2, is found to be stabilized in conditions when H3.3K27M is expressed, suggesting sequestration of Ezh2. Differently the frequency of binding of CBX7, the chromodomain protein of PRC1 and a "reader" of the of H3 modifications is reduced in H3.3K27M expressing cells, in agreement with the general reduction in H3.3K27me3 in these conditions. The part of the manuscript relating to Ezh2, aims to resolve conflicting biochemical data on the role of H3.3Me mutant in PRC2 targeting, and I feel these novel results would be very interesting for the researchers working on the epigenetics of pediatric gliomas. Differently, the results obtained on CBX7 are of little surprise as the authors have shown previously (Zhen et al. Elife, 2016), that PRC1 targeting depends on the recognition of H3.3K27Me3, and this modification is greatly reduced in cells carrying a copy of mutated H3.3. Nevertheless, the relevance of this mutation in pediatric cancer renders these results of potential great interest for the field. More globally, the paper is a first example about the role of how epigenetic alterations in cancer can divert the search mechanism of chromatin modifiers, extending the scope of action of intranuclear single-molecule approaches that so far mostly focused on transcription factors. Therefore, I believe that the paper is a good candidate for publication in Nature Communications. However, I have some concerns on how bound fractions and search times are calculated by the authors and in my opinion these biases might undermine the interpretation of the results (see below). Also I have some concerns/comments on other technical and biological details that I would like the authors to address.

We are grateful to the Reviewer for their constructive comments on the manuscript. We have tried to address these concerns as detailed below by performing additional experiments and re-analyzing all the data.

MAJOR CONCERN: ANALYSIS OF THE DISTRIBUTION OF DIFFUSION COEFFICIENTS and SEARCH TIME CALCULATION.

2.2) If I understood right, the D_m are calculation is track based (it is basically extracted from the first point of the time-averaged mean squared displacement curve), and each measured diffusion coefficient is not weighted for the duration of the track. I believe that it would be fine to use this method if the authors could follow molecules in the whole 3D volume of the cell. However, due to the limited thickness of the observation slice, I believe that that the method used can result in strong underestimation of the bound fractions, with a bias that depends on the duration of the binding events. To explain this I have to use an example. First of all, let's start from the definition of bound fraction that the authors also use at page 20, line 511: $F_1 = \tau_{\text{average}} / (\tau_{\text{average}} + \tau_{3d})$. This is analogous (for ergodicity) to define F_1 as the fraction of molecules being bound at every SINGLE TIME. Let's now consider the case in which a molecule binds for a very long time, and stays unbound for the same amount of time (so it's bound fraction should be 50%), and let's assume that you are in a condition in which in your detection slice only two molecules are visible at each time. The bound molecule will always be the same one through-out the whole movie and therefore will contribute with only one count to the distribution of D_m . Instead the free molecules will be in constant exchange, for every molecule that leaves the detection slice (typically in a couple of frames), a second one will jump in (the system is at equilibrium) and therefore the free molecules will contribute with MANY counts on the histogram of D_m , EVEN IF at every single time you just have one free and one bound molecule in the image. The problem is very evident for the H2A movie the authors provide as SI. In each frame most of the molecules (I would say more than 90%) appear as bound with only one or two

molecules diffusing around. Therefore one would expect to measure a bound fraction higher than 90%. However, due to the fact that bound molecules are counted only once, independently on the number of frames for which the molecules appear as bound, the bound fraction estimated by the author's method is much lower (let's than 70%). Importantly, I believe that the alternative way of analyzing the data proposed by the authors in the supplementary information (looking at just the first displacement of each track) would be affected by the same bias, and therefore an alternative strategy should be implemented to correctly analyze bound fractions. One trivial way to correct this issue would be to state that the measured bound fractions represent the fraction of TRACKS (rather than molecules) in the bound state. Unfortunately this approach would fail for the measurements of search times, as this calculation requires to measure the TRUE bound fraction (that is the fractional time a molecule stays in the bound state). Approaches that could be used to correctly calculate the bound fraction are the mathematical model derived in (Mazza et al, Nucl Ac Res, 2012), or the simplified methods used in either (Loffreda et al., Nat Comm, 2017) or (Hansen et al. eLife, 2017). Recently, Dr. Darzacq group has developed a web tool that allows to fit the tracking data with the last listed method (Hansen et al Biorxiv, 2017). In the paper describing the interface, the authors highlight the risks of misestimating bound fractions using a track-based approach in HILO SMT of nuclear factors. Given the little detail provided in how bound fractions are estimated. I might have missed something in how the authors analyzed their tracking data. In this case, I apologize for my comment, and I suggest the authors to better explain how they account for the different duration of the free and bound track segments.

We understood the concern raised by the Reviewer. Our LogD analysis is track-based. We agree that the LogD analysis underestimates the bound fraction. In the revised manuscript, we have re-analyzed all the data using Spot-On developed by Dr. Darzacq group (Hansen et al. Elife 7(2018)). The new analysis shows that the LogD analysis underestimates the bound fraction, which is consistent with the prediction by the Reviewer. Typically, the bound fraction obtained by the Spot-On analysis is increased by ~ 10% in comparison with the LogD analysis, from ~14% to ~22% for PRC2 and from ~30% to ~40% for Cbx7. This is also the case for H2A from ~67% to ~82% and for HaloTag-NLS from 0% to ~8%. The new analysis does not change the conclusions of the manuscript. In the revised manuscript, we have replaced all the LogD analysis with the displacement (Spot-On) analysis.

OTHER TECHNICAL COMMENTS:

2.3) 1. *The authors refer to long-lived and short-lived binding as the result of bi-exponential fitting of the 1-CDF distribution of residence times arising from interleaved single molecule imaging at low frame rates. Next, the parameters arising from the fit are used to compute search times according to (Loffreda et al Nat. Comm, 2017). However, these calculations underlie the assumption that the long-lived interactions are the one that are biologically relevant (the "specific binding" to cognate sites). In the transcription factors SMT literature these assumptions are typically tested by imaging mutants with impaired specific binding (Gebphardt et al., Nat Methods, 2013, Chen et al., Cell 2014 and so on). What is the evidence for PRC1 and PRC2 members that long-lived binding is representative of binding to cognate sites? I recall that some of these tests were performed on Cbx7 in their previous paper, but it would be useful to show that an incompetent Ezh2 has a reduced fraction of molecules engaged in long-lived binding. The EED KO experiment is somewhat arguing towards that direction, however to me it is puzzling that upon EED depletion, the search times of Ezh2 decrease and the authors should clarify the meaning of this result.*

We agree that in the original manuscript, the evidence that the long-lived interactions are specific binding events was weak. In the revised manuscript, we have performed additional experiments to support that the long-lived molecules are likely to be the specific binding ones. First, we made one Ezh2

mutant (Ezh2^{ΔSANT2}) that lacks the SANT2 domain that interacts with Suz12 (Justin et al. Nat Commun 7, 11316 (2016)). Previous studies showed that Suz12 is the key subunit that facilitates binding PRC2 to nucleosome (Nekrasov et al. EMBO Rep 6, 348-53 (2005)). The long-lived fraction and the residence time of Ezh2^{ΔSANT2} were greatly reduced in comparison with wild-type Ezh2 (Fig. 3 and Table S1). Second, HaloTag-NLS had a negligible long-lived fraction (Fig. 2B and Table S1). Third, the residence time and long-lived fraction of Ezh2^{Catalytic}, the catalytic lobe of Ezh2 (Justin et al. Nat Commun 7, 11316 (2016)), were greatly reduced compared to wild-type Ezh2 (Fig. 2, Fig. 6, and Table S1). Finally, in the original manuscript, we showed that the residence time of long-lived Ezh2 in *Eed* KO mES cells was greatly reduced compared to wild-type mES cells (Fig. 3 and Table S1). In summary, we suggest that the long-lived Ezh2 molecules are likely to be the specific binding ones.

We thank the Reviewer for pointing out that the meaning of search time was not clear. In the revised manuscript, we derived the kinetic relationship between the long-lived fraction, the search time, and the residence time as follows (p. 20-21).

$$\begin{aligned}
 F_{1sb} &= \frac{S}{F + N + S} \\
 &= \frac{k_{on}^{sb*} \times k_{off}^{tb}}{k_{off}^{tb} \times k_{off}^{sb} + k_{on}^{tb*} \times k_{off}^{sb} + k_{on}^{sb*} \times k_{off}^{tb}} \\
 &= \frac{(k_{on}^{sb*} \times k_{off}^{tb}) \times \frac{1}{k_{on}^{sb*} + k_{on}^{tb*}} \times \frac{1}{k_{off}^{sb} \times k_{off}^{tb}}}{(k_{off}^{tb} \times k_{off}^{sb} + k_{on}^{tb*} \times k_{off}^{sb} + k_{on}^{sb*} \times k_{off}^{tb}) \times \frac{1}{k_{on}^{sb*} + k_{on}^{tb*}} \times \frac{1}{k_{off}^{sb} \times k_{off}^{tb}}} \\
 &= \frac{\frac{f_{1sb}}{k_{off}^{sb}}}{\frac{1}{k_{on}^{tb*} + k_{on}^{sb*}} + \frac{1 - f_{1sb}}{k_{off}^{tb}} + \frac{f_{1sb}}{k_{off}^{sb}}} \\
 &= \frac{\tau_{sb}}{N_{trial} \times \tau_{3D} + (N_{trial} - 1) \times \tau_{tb} + \tau_{sb}} \\
 &= \frac{\tau_{sb}}{\tau_{search} + \tau_{sb}}
 \end{aligned}$$

The long-lived fraction is determined by the search time and the residence time. *Eed* knockout had no effect on the long-lived fraction of Ezh2, but reduced its residence time. Thus, the search time was reduced. Previous studies have shown that *Eed* is essential for the H3K27 methyltransferase activity *in vivo* and *in vitro* (Cao et al. Mol Cell 15, 57-67 (2004) and Montgomery et al. Curr Biol 15, 942-7 (2005)). Our results suggest that the number of Ezh2 molecules on chromatin and its target search time may not be a good predictor for the PRC2 activity *in vivo*, since *Eed* knockout has no effect on the long-lived fraction of Ezh2 and shortens the target search process of Ezh2. We propose that the residence time of Ezh2 on chromatin may be a better prediction for the methyltransferase activity of PRC2. Please see p. 13-14 line 348-362 for a full discussion of the relationship among the residence time, the search time, the bound fraction, and the PRC2 activity.

2.4) 2. *The authors use quite long exposure times (30ms) to track molecules, and in this situation motion blur might affect the estimates of BOTH the fractions of bound/free molecules and the estimation of the diffusion coefficients itself(as fast diffusing molecules will be systematically more affected by motion blur compared to slow ones). The effect has been described in several papers (Izeddin et al., eLife, 2014; Hansen et al, eLife, 2017). In the methods sections, the authors recognize this potential issue, however, in*

my opinion they underestimate the effect that such bias could have on their measurements. As the estimation of the diffusion coefficients might be affected by motion blur as well, different proteins might have different REAL diffusion coefficients, and therefore the bias favoring bound molecules might have different magnitudes in the different conditions tested. While the effect might not be dramatic, it still needs to be tested. Typically, motion blur is minimized by using shorter laser exposure times (< 5 ms), potentially via stroboscopic illumination. I suggest the authors to try this approach in a few key conditions (e.g. Cbx7 in WT and H3.3K27M expressing cells), to cross-validate their results. Also, for the problems mentioned above, currently the discussion on the differences in diffusion coefficients (for example lines 109-113 in the results) could be misleading.

We agree that using stroboscopic illumination with short exposure time can reduce the effects of motion blur on the estimation of bound fraction and diffusion constants. To investigate whether motion blur affects our conclusions, we have performed a series of experiments (Ezh2 and Cbx7 in H3.3- and H3.3K27M-expressing cells; and Ezh2 and Cbx7 in 9427 and SF8628 cells) by varying exposure times 5 ms, 10 ms, 20 ms, and 30 ms *via* stroboscopic illumination (Fig S4-5). These results indicate that motion blur results in a slight overestimation of bound fraction (Table S1), but does not affect our conclusions. We have updated the text as follows (p. 19 line 500-508).

“Another bias is that 3D freely diffusing molecules tend to “motion blur” because they can move several pixels during a short exposure time. The extent of the bias by motion blur is sensitive to the experimental conditions and the localization algorithm (Hansen et al. Elife 7(2018)). To test whether motion blur influences our conclusions, we performed a series of experiments in which we fixed the lag time (30 ms), but varied the exposure times (5 ms, 10 ms, 20 ms and 30 ms) (Fig S4-5). Motion blur caused a slight overestimation of bound PcG proteins (Table S1), but did not affect our conclusion. It should be noted that even at less than 5 ms exposure time, motion blur still contributes to the overestimation of bound PcG proteins. However, our optic setup limits us to further reduce the exposure time due to signal-to-noise.”

GENERAL COMMENTS ON THE BIOLOGICAL QUESTIONS

2.5) 1. Sequestration of PRC2. *As the authors report in the introduction, there is an unresolved issue in the literature as some reports show enrichment of PRC2 on H3K27M containing nucleosomes, while more recent studies show no evidence of PRC sequestration at these sites by ChIP-seq. The in-vitro single molecule data of the authors provide compelling evidence that some sequestration occurs. Similarly, the in-vivo SMT data shows that on average Ezh2 binding lasts 1.5× longer when H3K27M is expressed, albeit it is currently not clear if this longer binding occurs at loci enriched for this mutated histone. I feel that in the current form of the manuscript it is still not clear whether the authors believe that PRC2 is sequestered in cis- at H3K27M sites or if the epigenetic reprogramming caused by H3K27M is responsible for diverting PRC2 search. The authors could try to restate the relevant parts of the discussion to better clarify how their data fit in the current models for PRC2 inhibition in pediatric glioma. Ideally some additional data could help in clarifying the underlying mechanism.*

We appreciate the insightful and valuable comments from the Reviewer. We agree that in the original manuscript we have not provided clear mechanisms that underpin how H3K27M inhibits the trimethylation activity of PRC2 *in vitro* and reprograms the global occupancy of H3K27me3 and PRC2 *in vivo*. We have performed additional experiments and re-analyzed all the data to address these critiques. We have updated the discussion accordingly. We briefly discuss the mechanisms as follows. Please see p. 12-14 for a full discussion of mechanistic insights.

***In vivo* mechanisms underlying the loss/retainment of H3K27me3**

H3K27M results in a global reduction in the H3K27me3 level, but selectively retains a subset of loci with H3K27me3 and PRC2 (Chan et al. *Genes Dev* 27, 985-90 (2013) and Mohammad et al. *Nat Med* 23, 483-492 (2017)). Our analysis indicates that H3.3K27M increases the τ_{sb} of Ezh2 from ~10 s to ~15 s and the τ_{search} of Ezh2 from ~200 s to ~280 s. How does this explain the altered genomic occupancy of H3K27me3 and PRC2 by H3K27M? The interval (s) of sampling of specific sites has been described as $\frac{(\tau_{sb} + \tau_{search}) \times \text{The number of specific sites}}{\text{The number of PRC2 molecules in the nucleus}}$ (Chen et al. *Cell* 156, 1274-1285 (2014)). Our single-molecule data provide evidence that H3.3K27M prolongs not only the residence time of PRC2, but also its search time. The search time of Ezh2 is ~20-fold longer than its residence time. Previous report showed that the PRC2 level in *H3.3K27M*-expressing cells is the same as that in *H3.3*-expressing cells (Chan et al. *Genes Dev* 27, 985-90 (2013)). If we assume that the number of specific sites in *H3.3*- and *H3.3K27M*-expressing cells remains unchanged, the search time of PRC2 rather than its residence time contributes greatly to its sampling frequency. Therefore, the lengthened search time increases the interval of sampling of specific PcG-targeted sites. The reduced sampling frequency has differential effects on Polycomb targets: less effects on strong Polycomb targets (more binding sites), but greater effects on weaker Polycomb targets (less binding sites). This model agrees with a recent study that demonstrated strong Polycomb targets have clusters or long CpG islands (more binding sites) and high PRC2/H3K27me3 enrichment, while weak Polycomb target sites have isolated or short CpG islands (less binding sites) and low PRC2/H3K27me3 enrichment (Mohammad et al. *Nat Med* 23, 483-492 (2017)). Weak Polycomb targets tend to lose H3K27me3 while strong Polycomb targets are more likely to retain H3K27me3 (Mohammad et al. *Nat Med* 23, 483-492 (2017)). Thus, our live-cell single-molecule results provide insights into the mechanism that underpins how the reduced sampling frequency of PRC2 reduces the global H3K27me3 level, but retains H3K27me3 at some loci.

***In vitro* mechanisms underlying the inhibition of trimethylation activity**

H3K27M-nucleosomes are remarkably potent inhibitors of PRC2 (Lewis et al. *Science* 340, 857-61 (2013) and Brown et al. *J Am Chem Soc* 136, 13498-501 (2014)). However, a recent study using nucleosomes as substrates showed that the H3K27M mutation has minor effects on PRC2-nucleosome binding (Wang et al. *Nat Struct Mol Biol* 24, 1028-1038 (2017)). How do our single-molecule results reconcile with the seemingly controversial results? In these binding experiments (Wang et al. *Nat Struct Mol Biol* 24, 1028-1038 (2017)), the binding affinity was characterized by the equilibrium dissociation constant (K_D), a ratio of off-rate versus on-rate. Our *in vivo* and *in vitro* single-molecule analysis demonstrates that the H3K27M oncogenic mutation prolongs the residence time of Ezh2 on H3K27M-nucleosome. The lengthened residence time results in a reduction in the effective concentration of PRC2 by sequestering PRC2 on H3K27M-nucleosome, which leads to a reduced reaction rate of the trimethylation of H3K27 by PRC2 *in vitro*.

2.6) Did the authors tried to use a fluorescent version of H3K27M (for example by tagging it with GFP) to show whether this mutant localizes in discrete foci in the nucleus? If yes, the authors could try to measure whether the residence time of Ezh2 is indeed longer at these foci compared to the rest of the nucleoplasm using a single molecule approach such as the one previously described in (Morisaki et al., *Nat. Comm.*, 2014).

We have not tried to tag H3.3K27M with fluorescent proteins. However, we have expressed H3.3K27M-FLAG in HEK293T cells. Immunostaining of H3.3K27M-FLAG by anti-FLAG antibody showed that H3.3K27M-FLAG forms relatively uniform distribution in the nucleus (Fig. 4A).

2.7) *For the "global reprogramming" hypothesis: is there any ChIP-Seq data showing that the positions of PRC2 binding sites are widely different in cells expressing H3.3K27M compared to WT? If not, would it be possible to generate these kinds of data? I am aware that these experiments might extend beyond the scope of the paper. The authors should therefore, at minimum, restate the discussion to specify that future experiments are needed to clarify the mechanism underlying the control of PRC2 search by H3.3K27Me.*

We agree with the Reviewer that it would be of great interest to detect the positions of PRC2 binding sites and H3.3K27M-nucleosomes in *H3K27M*- and *H3*-expressing cells. Kristian Helin's group has performed ChIP-Seq to generate such data (Mohammad et al. Nat Med 23, 483-492 (2017)). Their results showed that both H3K27me3 and PRC2 subunit Suz12 are reprogrammed when H3K27M is expressed. Their results also showed that strong Polycomb targets have clusters or long CpG islands (more binding sites) and high PRC2/H3K27me3 enrichment, and that weak Polycomb target sites have isolated or short CpG islands (less binding sites) and low PRC2/H3K27me3 enrichment (Mohammad et al. Nat Med 23, 483-492 (2017)). Weak Polycomb targets tend to lose PRC2/H3K27me3 while strong Polycomb targets are more likely to retain PRC2/H3K27me3 (Mohammad et al. Nat Med 23, 483-492 (2017)). These results support our sampling model (p. 12-13 line 310-340).

2.8) *CBX7 over-expression and DIPG cells proliferation. The results of Figure 6 are interesting although not completely novel, as (Yu et al., Oncotarget, 2017) have recently shown that Cbx7 is a prognostic marker for gliomas and its overexpression result in reduced proliferation. This paper also show a hint about the mechanism by which Cbx7 overexpression could promote cell cycle arrest, by binding and repressing the Cyclin E1 promoter. As presented, the results do not clarify how the decrease in proliferation relates to the search mechanism of CBX7. At minimum, the authors should show the effect of tuning the expression levels of CBX7 on its search mechanism. Ideally the CBX7 search time should remain unchanged or (even better) decrease. Either of these two situations would mean that any given time more target sites are bound by CBX7 when over expressed. This observation could be then reconciled with the data from Yu et al.*

We are grateful to the Reviewer for their suggestion. We performed additional experiments to understand how increasing Cbx7 levels influence its binding and search mechanism. Surprisingly, our single-molecule analysis shows that increasing Cbx7 protein levels has no effect on the number of long-lived Cbx7 molecules on specific sites, but prolongs its residence time and search time (Fig. 8). The increased search time is due to the fact that Cbx7 needs more trials to find specific sites since more Cbx7 molecules non-specifically bind to chromatin (Table S1). We propose that the prolonged residence time facilitates recruiting other co-repressors by Cbx7, thereby repressing gene transcription. Consistently, emerging evidence suggests that the residence time of transcription factors directly correlates with transcriptional output (Loffreda et al. Nat Commun 8, 313 (2017), Choi et al. Nat Struct Mol Biol 24, 1039-1047 (2017), Stavreva et al. Molecular and Cellular Biology 24, 2682-2697 (2004), Lickwar et al. Nature 484, 251-U141 (2012), and Clauss et al. Nucleic Acids Research 45, 11121-11130 (2017)). Please see p. 11 line 279-292 for a full description of Results and p. 14 line 372-379 for a full discussion of the mechanistic insights.

TO SUM UP:

2.9) *The authors show interesting modulation of PcG protein search mechanisms depending on the expression of mutated histones. I have some concerns on the way the SMT data has been analyzed, and if the main results hold true after further validation, the paper is in my opinion of high interest. However,*

the authors should better define how Cbx7 overexpression affects its search mechanism and to clarify their model for PRC2 inactivation in cells expressing H3K27M.

We appreciate the insightful comments provided by the Reviewer. We have re-analyzed all the data using Spot-On. The conclusions still hold true. By performing additional experiments and re-analyzing all the data, we define how Cbx7 levels affect its binding and search mechanism and clarify the molecular mechanisms that underpin how H3K27M inhibits the trimethylation activity of PRC2 *in vitro* and how H3K27M reprograms the genomic occupancy of H3K27me3 and PRC2 *in vivo*.

We would like to thank the Reviewer again. The comments raised by the Reviewer prompted us to re-analyze the causal relationship between the *in vivo* kinetic parameters (the fraction of PcG protein on chromatin, the stability of PcG protein on chromatin (residence time), and the time for finding a specific PcG site (search time)), the PcG activity and the PcG occupancy. Based on a large number of *in vivo* single-molecule data from 18 cell lines and literature, we now propose that the residence time (the stability) of long-lived PcG molecules directly correlates with their functions and the search time plays a critical role in regulating the global occupancy of PcG proteins.

Minor Points:

2.10) -p. 10 Last sentence of results. “Proliferation” should be changed in “proliferating”

We revised the original sentence as “These results demonstrate that human DIPG cells grow slowly in response to increasing Cbx7 expression” (p. 11 line 274).

2.11) - p. 16 Line 401. The range of dye concentration used should be specified.

The dye concentrations have been specified (p. 17 line 455).

2.12) - P. 16. Line 418. The laser power is expressed in mW. However it would be more useful to provide the irradiance, given by the laser power divided by the area of the field of view illuminated by the laser – so that others could try to replicate the settings.

The laser power density has been provided in the revised manuscript (p. 17 line 472 and p. 23 line 607).

Reviewer #3 (Remarks to the Author):

3.1) *The Histone H3 K27M missense mutation is a frequently found genetic alternation in pediatric diffuse intrinsic pontine glioma (DIPG) and other midline high-grade gliomas. The K27M mutant histone leads to a loss of H3K27me2/3 apparently via inhibition of the polycomb repressive complex 2 (PRC2). Despite global loss of K27me2/3, focal gains of K27me3 were noted in K27M-containing cell lines. How these K27me3 foci contributed to K27M-mediated tumorigenesis remains unknown.*

We thank the Reviewer for their criticisms that help us improve the manuscript. To provide the molecular mechanisms that underpin how H3K27M inhibits the trimethylation activity of PRC2 *in vitro* and reprograms the genomic occupancy of H3K27me3 and PRC2 *in vivo*, we have performed additional experiments and re-analyzed all the data. We have re-written most of the Results section and the entire Discussion section accordingly.

3.2) *Here, the authors used a live-cell single molecule tracking system to explore diffusion of PRC1 (CBX7) and PRC2 (EZH2 and EED) subunits in murine embryonic stem cells. Based on SMT measurements, the authors conclude that different populations (chromatin-bound, confined diffusion, free diffusion) exist for PRC1 and PRC2 complexes. Residence time of chromatin-bound complexes was also assessed. The authors find that cells expressing the K27M histone alter the diffusion kinetics of CBX7, but not PRC2 subunits in HEK293T cells. While the diffusion kinetics of PRC2 was not affected in K27M-expressing HEK293T cells, oddly, the authors did find that K27M affected the diffusion of both EZH2 and CBX7 in DIPG cells. In support of findings from previous studies, the authors find that K27M stabilizes EZH2 on nucleosomes in vitro. The authors find that overexpression of CBX7 suppresses DIPG cell growth in vitro.*

We apologize that we did not make clear about some results in the original manuscript. In the revised manuscript, we have re-written most of the Results section. We summarize the main results from both original and new data as follows.

- (1) PRC2 and Cbx7 have a similar residence time on chromatin in mES cells, but exhibit different fractions bound to chromatin and distinct search times, revealing that their search processes determine their levels bound to chromatin.
- (2) *Eed* knockout destabilizes *Ezh2* on chromatin and accelerates the search process of *Ezh2*, thereby having no effect on its chromatin-bound level. The disruption of interactions between *Ezh2* and *Suz12* destabilizes *Ezh2* on chromatin and prolongs the search process of *Ezh2*, thereby reducing its fraction bound to chromatin. These results suggest that the bound level and the search time may not be a good prediction for the activity of PRC2 and that the residence time directly correlates with the PRC2 activity.
- (3) Our data from both HEK293T cells and patient-derived DIPG cells show that H3.3K27M prolongs the search time and residence time of *Ezh2*, but has no effect on its fraction bound to chromatin. The search time of *Ezh2* is ~20-fold longer than its residence time, arguing that the prolonged search time reduces the sampling frequency of PRC2 for specific sites, which leads to the loss and retainment of H3K27me3 *in vivo*.
- (4) Our data from both HEK293T cells and patient-derived DIPG cells demonstrate that H3.3K27M has no effect on the stability of *Cbx7* on chromatin, but prolongs its search process and reduces its fraction bound to chromatin, indicating that the prolonged search time reduces its fraction bound to chromatin.
- (5) Increasing expression of *Cbx7* inhibits the proliferation of patient-derived DIPG cells and prolongs its residence time, but has no effect on the number of long-lived *Cbx7* molecules on the specific sites, arguing that the residence time of *Cbx7* directly correlates with its function.

Our analyses from 18 cell lines suggest that the stability/residence time of PcG proteins on chromatin directly correlates with their functions and their search time plays a critical role for regulating their genomic occupancy.

3.3) *The observation that the K27M mutation affects PRC2 and PRC1 localization has been noted in previous studies using high resolution ChIP sequencing techniques. The observation that overexpression of CBX7 suppresses growth of DIPG cells is potentially interesting if the mechanism could be teased out. Overall, the manuscript has some interesting observations, but lacks mechanistic insight into how the presence of K27M histones leads to an increase in PRC2 activity at specific genic loci. Also, it does little to explain how a mutant histone leads to a unique chromatin state that promotes tumorigenesis.*

We understood the critiques raised by the Reviewer. We have performed additional experiments and re-analyzed all the data to address these critiques. Based upon classical *in vitro* biochemical measurements and genome-wide studies, two models (the sequestration model (Justin et al. Nat Commun 7, 11316 (2016)) and the inhibition model (Mohammad et al. Nat Med 23, 483-492 (2017))) have been proposed to explain the loss/retainment of H3K27me3 and PRC2 *in vivo*. The two models deepen our understanding of the loss/retainment of H3K27me3 and PRC2. However, a recent genome-wide study showed that H3K27M localizes to transcriptionally active chromatin regions and at gene regulatory elements and PRC2 subunits Ezh2 and Suz12 are largely excluded from chromatin on sites containing H3K27M (Piunti et al. Nat Med 23, 493-500 (2017)). Another recent study using nucleosomes as substrates showed that the H3K27M mutation has minor effects on PRC2-nucleosome binding *in vitro* (Wang et al. Nat Struct Mol Biol 24, 1028-1038 (2017)). These studies argue that the mechanisms that underpin how H3K27M inhibits the trimethylation activity of PRC2 *in vitro* and epigenetically reprograms H3K27me3/PRC2 *in vivo* remain incompletely understood. One of the unique features of our manuscript is that we measure directly the residence time and search time of PcG proteins, as well as their bound fraction in living cells. The residence time and search time, which are analogous to the inverse of off-rate and on-rate described in classical kinetics, are the foundation for understanding of how nuclear factors find/search for their targets and perform their functions on chromatin *in vivo*. We found that PRC2 proteins take ~200 s to find their target sites, while stay on their target sites for ~10 s, immediately suggesting that the search time plays a critical role in controlling the occupancy of PRC2. Our data also imply that the residence time directly correlates with the functions of PcG proteins. These results provide new perspectives on the dynamic processes of PcG proteins and the effects of H3.3K27M on the dynamic processes. Please see p. 12-14 for a full discussion of mechanistic insights. We summarize the three main mechanistic insights as follows.

Mechanism I: *The inhibition of trimethylation of H3K27 by PRC2 in vitro is through sequestering PRC2 by prolonging its residence time.*

H3K27M-nucleosomes are remarkably potent inhibitors of PRC2 (Lewis et al. Science 340, 857-61 (2013) and Brown et al. J Am Chem Soc 136, 13498-501 (2014)). However, a recent study using nucleosomes as substrates showed that the H3K27M mutation has minor effects on PRC2-nucleosome binding (Wang et al. Nat Struct Mol Biol 24, 1028-1038 (2017)). How do our single-molecule results reconcile with the seemingly controversial results? In these binding experiments (Wang et al. Nat Struct Mol Biol 24, 1028-1038 (2017)), the binding affinity was characterized by the equilibrium dissociation constant (K_D), a ratio of off-rate versus on-rate. Our *in vivo* and *in vitro* single-molecule analysis demonstrates that the H3K27M oncogenic mutation prolongs the residence time of Ezh2 on H3K27M-nucleosome. The lengthened residence time results in a reduction in the effective concentration of PRC2 by sequestering PRC2 on H3K27M-nucleosome, which leads to a reduced reaction rate of the trimethylation of H3K27 by PRC2 *in vitro*.

Mechanism II: *The loss/retainment of H3K27me3 in vivo is via a reduced sampling frequency of PRC2 by prolonging its search time.*

H3K27M results in a global reduction in the H3K27me3 level, but selectively retains a subset of loci with H3K27me3 and PRC2 (Chan et al. *Genes Dev* 27, 985-90 (2013) and Mohammad et al. *Nat Med* 23, 483-492 (2017)). Our analysis indicates that H3.3K27M increases the τ_{sb} of Ezh2 from ~10 s to ~15 s and the τ_{search} of Ezh2 from ~200 s to ~280 s. How does this explain the altered genomic occupancy of H3K27me3 and PRC2 by H3K27M? The interval (s) of sampling of specific sites has been described as $\frac{(\tau_{sb} + \tau_{search}) \times \text{The number of specific sites}}{\text{The number of PRC2 molecules in the nucleus}}$ (Chen et al. *Cell* 156, 1274-1285 (2014)). Our single-molecule data provide evidence that H3.3K27M prolongs not only the residence time of PRC2, but also its search time. The search time of Ezh2 is ~20-fold longer than its residence time. Previous report showed that the PRC2 level in *H3.3K27M*-expressing cells is the same as that in *H3.3*-expressing cells (Chan et al. *Genes Dev* 27, 985-90 (2013)). If we assume that the number of specific sites in *H3.3*- and *H3.3K27M*-expressing cells remains unchanged, the search time of PRC2 rather than its residence time contributes greatly to its sampling frequency. Therefore, the lengthened search time increases the interval of sampling of specific PcG-targeted sites. The reduced sampling frequency has differential effects on Polycomb targets: less effects on strong Polycomb targets (more binding sites), but greater effects on weaker Polycomb targets (less binding sites). This model agrees with a recent study that demonstrated strong Polycomb targets have clusters or long CpG islands (more binding sites) and high PRC2/H3K27me3 enrichment, while weak Polycomb target sites have isolated or short CpG islands (less binding sites) and low PRC2/H3K27me3 enrichment (Mohammad et al. *Nat Med* 23, 483-492 (2017)). Weak Polycomb targets tend to lose H3K27me3 while strong Polycomb targets are more likely to retain H3K27me3 (Mohammad et al. *Nat Med* 23, 483-492 (2017)). Thus, our live-cell single-molecule results provide insights into the mechanism that underpins how the reduced sampling frequency of PRC2 reduces the global H3K27me3 level, but retains H3K27me3 at some loci.

Mechanism III: *Increasing Cbx7 concentrations regulates its binding and search dynamics.*

Our data show that increasing expression of Cbx7 inhibits the proliferation of DIPG cells. A recent study demonstrated that overexpression of Cbx7 arrests glioma cells in the G_0/G_1 phase (Yu et al. *Oncotarget* 8, 26637-26647 (2017)). It has been well known that the activity of many nuclear factors is regulated by modulating their concentration in the nucleus. Our single-molecule analysis demonstrates that increasing Cbx7 protein levels have no effect on the number of long-lived Cbx7 molecules on the specific sites, but instead prolong its residence time. We propose that the prolonged residence time facilitates recruiting other co-repressors by Cbx7, thereby repressing gene transcription. Consistently, emerging evidence suggests that the residence time of transcription factors directly correlates with transcriptional output (Loffreda et al. *Nat Commun* 8, 313 (2017), Choi et al. *Nat Struct Mol Biol* 24, 1039-1047 (2017), Stavreva et al. *Molecular and Cellular Biology* 24, 2682-2697 (2004), Lickwar et al. *Nature* 484, 251-U141 (2012), and Clauss et al. *Nucleic Acids Research* 45, 11121-11130 (2017)).

Additionally, we provided a few other insights into the roles of the residence time and target search time in the activity and genomic occupancy of PRC2 and Cbx7 (p. 12-14).

3.4) *The manuscript relies too much on only using the SMT technique and fails to use orthogonal approaches to support conclusions based on SMT data.*

We agree with the Reviewer that experiments ideally should be cross-validated. We would like to point out that the search process of transcription factor for specific DNA sites has been theoretically discussed

over the last 40 years (Berg et al. Biochemistry 20, 6929-48 (1981), Winter et al. Biochemistry 20, 6948-60 (1981), Winter et al. Biochemistry 20, 6961-77 (1981), and Halford et al. Biochem Soc Trans 37, 343-8 (2009)), but direct measurements particularly in mammalian cells have just begun a few years ago (Izeddin et al. Elife 3(2014), Chen et al. Cell 156, 1274-1285 (2014), and Loffreda et al. Nat Commun 8, 313 (2017)), which is primarily due to lack of technologies and approaches. To our best knowledge, the search kinetics of epigenetic regulatory complexes for specific sites has not been reported *in vivo* and the state-of-the-art live-cell single-molecule tracking is the only technique that is currently available for addressing their search processes *in vivo*. The search and binding kinetics of epigenetic complexes in living cells are the fundamental basis for our understanding of transcriptional dynamics.

3.5) *In summary, in the absence of any new mechanistic data about PRC2, or how K27M leads to gain/loss of K27me3, I believe that this manuscript does not provide sufficient novel information to justify publication in Nature Communications.*

We would like to thank the Reviewer again. By re-analyzing all the data and performing additional experiments, we provide insights into the inhibition of trimethylation activity of PRC2 *in vitro* and the loss/retainment of H3K27me3 *in vivo*. Our analyses suggest that the reduced sampling frequency of PRC2 by prolonging the search time for target sites reduces the global H3K27me3 level, but retains H3K27me3/PRC2 at some loci. These results highlight the importance of directly measuring the kinetic parameters (search time, residence time, and chromatin-bound level) of PcG proteins as they perform essential time-dependent activities in living cells. These essential measurements of PcG search and binding modes provide new perspectives on the dynamic behavior of PcG-chromatin transactions that are the basis of their genomic occupancy and activity.

In summary, our manuscript represents the first example in which a cancer mutation influences the binding and search process of epigenetic complexes, thereby affecting their activity and genomic occupancy. Our data highlight the importance of residence time of PcG proteins in their activity and the critical role of search time of PcG proteins in their genomic occupancy. These innovative concepts have been incorporated into the revised manuscript.

Other comments:

3.6) *Need to define the term 'SMT' in the paper. The term is presented without explanation.*

We have defined the term 'SMT' in the paper.

3.7) *Line 110 "EZH2 and Eed dynamically exchange"-- too strong without additional data supporting this finding. Include "may or might dynamically exchange".*

Thanks for carefully reading the manuscript. We have revised the sentence.

Reviewers' Comments:

Reviewer #1 (Remarks to the Author):

Overall assessment

The revised paper from Tatavosian, Huynh & Duc from Xiaojun Ren's laboratory is a substantial improvement. I am very impressed with the revision and the authors have clearly taken the reviewer comments very seriously and gone out of their way to address them. In my previous review, I mainly raised concerns about the SMT for measuring diffusion constants and bound fractions and the authors have fully addressed my concerns and I believe the present manuscript is technically solid in this regard and above and beyond the standard in the field.

I would like to mention that the authors now explicitly tested the effect of "motion blur" in Suppl. Figure S4, which shows a small effect and gives the reader confidence in the results and their updated discussion in the methods is clear, comprehensive and honest. This is great.

I would also like to highlight that the number of different techniques, cell lines and genetic engineering approaches used is really quite impressive. The perturbation experiments in particular are quite informative and it is a pleasure to see such a wide range of approaches all directed at a biologically important problem in H3.3K27M.

In summary, I believe the revised manuscript is technically sound, quite impressive and likely to be of wide interest to both the Polycomb, cancer and imaging communities.

Minor comment

Figure 4: it seems a bit weird that "D" is above "B" in the figure panels.

Reviewer #2 (Remarks to the Author):

It is clear that the authors have taken very seriously the comments of the reviewers. The manuscript has clearly improved: the analysis of SMT data is much improved, the authors added several controls and many additional experiments. In my opinion, the manuscript provides important insights about the dynamic behavior of PRC complexes in living cells and I would recommend its publication in Nature Communications.

I have a couple of minor comments:

- Abstract: I find it not so well focused: The sentence "However, the underlying mechanisms remain elusive" is too general, and it should be more clear that the authors focus on the role of H3.3K27M mutation in affecting PcG protein kinetics and function.

- Discussion: Lines 341-346 Cbx7 increase its search time in H3.3K27M: The authors could state clearer that the increase of search time in these settings, might be simply due to the fact that the number of binding sites is decreased (less H3.3K27me3).

- Discussion: Line 318-325. The hypothesis that the decrease in sampling differentially affect strong Polycomb targets and weak ones, is very compelling, but I feel that it is still a little speculative at this stage. I suggest to clarify in the discussion that this is a speculation at this stage. Could the authors suggest future experiments to prove this model?

Reviewer #3 (Remarks to the Author):

Overall, the revised manuscript is improved over the initial submission. Many of the reviewer comments were addressed. Also, the discussion of the study's major findings within the context of the literature regarding the K27M mechanism is improved. I noted that the observation that overexpression of CBX7 suppresses growth of DIPG cells is interesting if further mechanistic insight can be made.

However, no additional experiments were conducted to address why over expression of a single subunit of the PRC1 complex (CBX7 in this case) could restore normal polycomb silencing. It's very surprising. But perhaps not so if the concentration of CBX7 rate-limiting to formation of the PRC1 complex in these cells. Is it? Do point mutations in the chromodomain that abolish interaction with K27me3 affect the growth suppressive activity of CBX7? Does expression of CBX7 restore polycomb-based silencing in the DIPG cells as assessed by gene expression? The overexpression still seems like preliminary result and perhaps a bit premature for publication. The mechanism remains unexplored and it doesn't really fit well with the rest of the manuscript.

Some minor comments regarding the discussion and previous studies:

1) A previous study published in a 2017 Biochemical and Biophysical Research Communications paper by Hetey and colleagues assessed the diffusion rate and found that EZH2 diffusion in live cells was not altered by the K27M histone H3. The authors should cite this paper and discuss its conclusions in relation to their own, especially since it's a highly relevant previous study.

2) Related to controversy regarding K27M interaction with PRC2 in vitro: It should be noted that the Wang et al. 2017 NSMB study did not assess binding of K27M to PRC2 in the presence of S-adenosyl methionine. Previously, it was noted that SAM stabilizes interaction between K-M mutant histones and SET domain methyltransferases (Justin et al. 2016 Nat. Commun, Jayaram et al. 2017 PNAS; Zhang et al. 2017 Sci Rep). There is little controversy related to the in vitro interaction between K27M and PRC2.

Responses to Reviewers' Comments:

Please note that the comments of the Reviewers are quoted verbatim using *italic and blue type*. Our rebuttal is in black.

REVIEWERS' COMMENTS:

Reviewer #1 (Remarks to the Author):

Overall assessment

The revised paper from Tatavosian, Huynh & Duc from Xiaojun Ren's laboratory is a substantial improvement. I am very impressed with the revision and the authors have clearly taken the reviewer comments very seriously and gone out of their way to address them. In my previous review, I mainly raised concerns about the SMT for measuring diffusion constants and bound fractions and the authors have fully addressed my concerns and I believe the present manuscript is technically solid in this regard and above and beyond the standard in the field.

I would like to mention that the authors now explicitly tested the effect of "motion blur" in Suppl. Figure S4, which shows a small effect and gives the reader confidence in the results and their updated discussion in the methods is clear, comprehensive and honest. This is great.

I would also like to highlight that the number of different techniques, cell lines and genetic engineering approaches used is really quite impressive. The perturbation experiments in particular are quite informative and it is a pleasure to see such a wide range of approaches all directed at a biologically important problem in H3.3K27M.

In summary, I believe the revised manuscript is technically sound, quite impressive and likely to be of wide interest to both the Polycomb, cancer and imaging communities.

Thanks.

Minor comment

Figure 4: it seems a bit weird that "D" is above "B" in the figure panels.

Thanks for carefully reviewing the manuscript. We have re-arranged the figure panels.

Reviewer #2 (Remarks to the Author):

It is clear that the authors have taken very seriously the comments of the reviewers. The manuscript has clearly improved: the analysis of SMT data is much improved, the authors added several controls and many additional experiments. In my opinion, the manuscript provides important insights about the dynamic behavior of PRC complexes in living cells and I would recommend its publication in Nature Communications.

Thanks.

I have a couple of minor comments:

- Abstract: I find it not so well focused: The sentence "However, the underlying mechanisms remain elusive" is too general, and it should be more clear that the authors focus on the role of H3.3K27M mutation in affecting PcG protein kinetics and function.

We agree with their comments. We have revised the original sentence as that "however, how the mutation affects kinetics and function of PcG proteins remain elusive."

- Discussion: Lines 341-346 Cbx7 increase its search time in H3.3K27M: The authors could state clearer that the increase of search time in these settings, might be simply due to the fact that the number of binding sites is decreased (less H3.3K27me3).

Thanks for the suggestion. We have made clearer statement by adding one sentence “The increase of search time of Cbx7 is due to the fact that the number of H3K27me3 is decreased.” (p14, line 353-354)

- Discussion: Line 318-325. The hypothesis that the decrease in sampling differentially affect strong Polycomb targets and weak ones, is very compelling, but I feel that it is still a little speculative at this stage. I suggest to clarify in the discussion that this is a speculation at this stage. Could the authors suggest future experiments to prove this model?

We understood this. We have revised the original sentence as that “We hypothesize that the reduced sampling frequency has differential effects on Polycomb targets....” (p13, line 332)

Given the recent advance in labelling specific genomic loci by CRISPR/dCas9 and single-molecule imaging, the sampling model could be further explored within living cells. Alternately, the sampling model could be tested in defined *in vitro* experimental systems using single-molecule imaging. Our laboratory is actively pursuing this direction.

Reviewer #3 (Remarks to the Author):

Overall, the revised manuscript is improved over the initial submission. Many of the reviewer comments were addressed. Also, the discussion of the study’s major findings within the context of the literature regarding the K27M mechanism is improved.

Thanks.

I noted that the observation that overexpression of CBX7 suppresses growth of DIPG cells is interesting if further mechanistic insight can be made. However, no additional experiments were conducted to address why over expression of a single subunit of the PRC1 complex (CBX7 in this case) could restore normal polycomb silencing.

We have shown previously that Cbx7-PRC1 targeting depends on the recognition of H3K27me3 (Zhen, et al. Elife 5(2016)). Given that this modification is greatly reduced in cells carrying H3.3K27M, the results about overexpression of Cbx7 suppressing growth of DIPG cells are compatible with the current model where H3K27me3 generated by PRC2 recruits Cbx7-PRC1.

In the revised manuscript, we have performed additional experiments to understand how increasing expression of Cbx7 alters its binding and search dynamics (Fig. 8). Classical kinetic studies suggest that transcription repression is often described by a repressor occupancy model (Hammar, et al. Nature Genet. 46(4), 405-408 (2014)). The repression ratio (RR) for a repression factors can be described as the following.

$$RR = \frac{\tau_{\text{search}} + \tau_{\text{sb}}}{\tau_{\text{search}}}$$

Furthermore, emerging data suggest that the residence time of transcription factors directly correlates with transcriptional output (Loffreda et al. Nat Commun 8, 313 (2017), Lickwar et al. Nature 484, 251-

U141 (2012), and Clauss et al. *Nucleic Acids Research* 45, 11121-11130 (2017)). Thus, it is crucial to measure the search and binding dynamics of transcription repressors.

Here, we show that increasing expression of *Cbx7* does not alter the number of *Cbx7* molecules on specific sites, but prolongs its residence time. We propose that the prolonged residence time facilitates recruiting other co-repressors by *Cbx7*, thereby repressing gene transcription.

It's very surprising. But perhaps not so if the concentration of CBX7 rate-limiting to formation of the PRC1 complex in these cells. Is it?

Our previous results demonstrated that the incorporation of *Cbx7* into the canonical PRC1 complexes inhibits binding of *Cbx7* to chromatin (Zhen, et al. *Elife* 5(2016)). Here, we show that increasing expression of *Cbx7* stabilizes *Cbx7* on chromatin. Thus, it is unlikely that the concentration of *Cbx7* is rate-limiting factor to form the PRC1 complex in these cells.

Do point mutations in the chromodomain that abolish interaction with K27me3 affect the growth suppressive activity of CBX7? Does expression of CBX7 restore polycomb-based silencing in the DIPG cells as assessed by gene expression?

We understood their comments. The questions raised by the reviewer are important and warrant further exploration. However, these questions are beyond the scope of the current manuscript. In the revised manuscript, we have added one sentence as follows.

“Further experiments are needed to test the casual relationship between residence time and transcriptional activity.” (p15, line 394-395)

The overexpression still seems like preliminary result and perhaps a bit premature for publication. The mechanism remains unexplored and it doesn't really fit well with the rest of the manuscript.

Our current manuscript focuses on the binding and target search dynamics of PcG proteins imposed by the DIPG H3.3K27M mutation. The target search and binding kinetics of chromatin-associating factors is the fundamental framework for understanding their genomic occupancy and activity. We have studied how increasing expression of *Cbx7* alters the binding and target search process of *Cbx7* in DIPG cells. Our data indicated that the residence time of *Cbx7* directly correlates with its inhibitory activity. Additionally, since H3.3K27M has opposite effects on the bound fraction and residence time of *Ezh2* and *Cbx7*, *Cbx7* serves as a contrast for our single-molecule system. Thus, we would like to state these studies are within the central theme of the manuscript: single-molecule dynamics of PcG proteins imposed by the DIPG H3.3K27M mutation.

Some minor comments regarding the discussion and previous studies:

1) A previous study published in a 2017 Biochemical and Biophysical Research Communications paper by Hetey and colleagues assessed the diffusion rate and found that EZH2 diffusion in live cells was not altered by the K27M histone H3. The authors should cite this paper and discuss its conclusions in relation to their own, especially since it's a highly relevant previous study.

Hetey et al in their BBRC paper measured the diffusion behavior of *Ezh2* in *H3.3*- and *H3.3K27M*-expressing cells by FRAP and FCS (Hetey, S. et al. *Biochem Biophys Res Commun* 490, 868-875 (2017)). Our live-cell single-molecule imaging not only measured diffusion coefficient, but also directly quantified residence time and target search time, which allows us to assess how the binding and target search

dynamics of PcG proteins affects their genomic occupancy and function. In the original manuscript, we tabled the diffusion constants in the Supplementary Information since these constants contribute less to our understanding of the binding and search processes of PcG proteins imposed by H3.3K27M. In the revised manuscript, we have added the following sentence.

“The diffusion coefficients of Ezh2 in *H3.3*- and *H3.3K27M*-expressing cells were similar, consistent with previous results measured from FRAP and FCS (Hetey, S. et al. *Biochem Biophys Res Commun* 490, 868-875 (2017))”. (p9, line 200-201)

2) Related to controversy regarding K27M interaction with PRC2 in vitro: It should be noted that the Wang et al. 2017 NSMB study did not assess binding of K27M to PRC2 in the presence of S-adenosyl methionine. Previously, it was noted that SAM stabilizes interaction between K-M mutant histones and SET domain methyltransferases (Justin et al. 2016 Nat. Commun, Jayaram et al. 2017 PNAS; Zhang et al. 2017 Sci Rep). There is little controversy related to the in vitro interaction between K27M and PRC2.

Since these experiments were carried out in different conditions, in our opinion, it is difficult to compare directly their conclusions. Thus, we have revised the text as follows.

“Studies have shown that H3K27M-nucleosomes are remarkably potent inhibitors of PRC2 and H3K27M peptides bind to the PRC2 active site with above 20-fold higher affinity than the equivalent peptides (Lewis, P.W. et al. *Science* 340, 857-61 (2013); Justin, N. et al. *Nat Commun* 7, 11316 (2016); and Brown, Z.Z. et al.. *J Am Chem Soc* 136, 13498-501 (2014)). However, a recent study using nucleosomes as substrates showed that the H3K27M mutation has minor effects on PRC2-nucleosome binding (Wang, X. et al. *Nat Struct Mol Biol* 24, 1028-1038 (2017)). The discrepancy could be due to the experimental conditions, such as the substrates used (peptides *versus* nucleosomes), the composition of PRC2 complexes, and the presence or absence of SAM.” (p15, line 374-379)